# Platonic Transformers: A Solid Choice For Equivariance

**Mohammad Mohaiminul Islam** [1]  **Rishabh Anand** [* 2]  **David R. Wessels** [* 3]  **Friso de Kruiff** [3]  **Thijs P. Kuipers** [4]
**Rex Ying** [2]  **Clara I. Sánchez** [1 4]  **Sharvaree Vadgama** [† 3]  **Georg Bökman** [† 3]  **Erik J. Bekkers** [3 5]

Open-source code: `github.com/niazoys/PlatonicTransformers`

## Abstract

While widespread, Transformers lack inductive biases for geometric symmetries common in science and computer vision. Existing equivariant methods often sacrifice the efficiency and flexibility that make Transformers so effective through complex, computationally intensive designs. We introduce the **Platonic Transformer** to resolve this trade-off. By defining attention relative to reference frames from the Platonic solid symmetry groups, our method induces a principled weight-sharing scheme. This enables combined equivariance to continuous translations and Platonic symmetries, while preserving the exact architecture and computational cost of a standard Transformer. Furthermore, we show that this attention is formally equivalent to a dynamic group convolution, which reveals that the model learns adaptive geometric filters and enables a *highly scalable, linear-time convolutional variant*. Across diverse benchmarks in computer vision (CIFAR-10), 3D point clouds (ScanObjectNN), and molecular dynamics, property prediction and generation (OMol25, ProteinMD, QM9), the Platonic Transformer achieves competitive performance by leveraging these geometric constraints at no additional cost.

## 1. Introduction

Transformers (Vaswani et al., 2017) have become widespread in deep learning, demonstrating unprecedented success on a massive scale (Dosovitskiy et al., 2021; Jumper et al., 2021; Devlin et al., 2019). Their power lies in simple, general-purpose mechanisms that have matured over the years and continue to offer remarkable gains in speed and flexibility, benefiting from vast datasets and computational resources. Yet, this very generality implies they are not inherently equipped to handle specific symmetries present in many scientific domains. For problems with geometric structure, such as those in physics, molecular chemistry, and 3D computer vision, performance can be significantly enhanced by incorporating such inductive bias (Fuchs et al., 2020; Ying et al., 2021; Zhao et al., 2021; Bekkers et al., 2024; Balla et al., 2024; Liao et al., 2024; Romero & Cordonnier, 2021; Wessels et al., 2024; Bose et al., 2024; Zhdanov et al., 2024; Nyholm et al., 2025). The principle of symmetry, for example, has given rise to highly data-efficient and robust group equivariant networks (Cohen & Welling, 2016; 2017; Cesa et al., 2022). However, scaling these symmetry-aware networks has been difficult, as their reliance on operations like group convolutions or Clebsch-Gordan tensor products introduces significant computational overhead compared to standard architectures (He et al., 2021a; Luo et al., 2024). This raises the question: *how can we leverage powerful geometric inductive biases within the transformer architecture without sacrificing the speed and flexibility integral to its success?*

A central challenge in addressing this problem lies in designing an attention mechanism that inherently respects geometric transformations. Such a mechanism would expand on the inductive bias of Transformers, which is typically limited to position embeddings. While widely used, absolute positional encodings provide location information, but they enforce no explicit relational structure (Shaw et al., 2018; He et al., 2021b). A significant step towards this goal has been the adoption of Rotary Position Embeddings (RoPE) (Su et al., 2024), which endows attention with translation equivariance. Yet, extending this to roto-translation equivariance within the standard Transformer framework remains challenging. Existing approaches often achieve this by making complex architectural changes to equivariant networks that poorly scale or settle for invariant attention mechanisms which sacrifice feature representations for sim-

---

[1]QurAI, University of Amsterdam, Netherlands [2]Yale University, USA [3]AMLab, University of Amsterdam, Netherlands [4]BMEP, Amsterdam UMC, Netherlands [5]New Theory. Correspondence to: Mohammad Mohaiminul Islam <niazoys94@gmail.com>.

*Proceedings of the 43rd International Conference on Machine Learning*, Seoul, South Korea. PMLR 306, 2026. Copyright 2026 by the author(s).

plicity and computational efficiency (Masters et al., 2022; Assaad et al., 2023; Thölke & Fabritiis, 2022; Brehmer et al., 2023; Kundu & Kondor, 2025; Joshi et al., 2025). Recent efforts have also explored *hybrid architectures* that resort to symmetry breaking (Qu & Krishnapriyan, 2024; Lawrence et al., 2025) to improve scalability but require a careful mix of modules to maximize downstream performance.

Our main contribution is the *Platonic Transformer*, a framework that achieves equivariance to continuous translations and discrete roto-reflections in Transformers *without changing the underlying attention mechanism or computation graph*. To achieve this, our method processes features relative to a collection of reference frames that form a Platonic symmetry group ($\mathcal{G} \subset O(3)$) and constrains all linear layers to be equivariant with respect to this choice of frame. This principled scheme allows the standard attention block, including its unmodified Rotary Position Embeddings (RoPE), to operate in parallel across these frames, and effectively associates each reference frame with a distinct attention head. As a result, the model incorporates a geometric inductive bias without altering the architecture or computational footprint of a standard Transformer. This enables flexible usage across domains at no additional cost, resolving the long-standing symmetry-awareness vs. scaling dilemma.

Additionally, we analyze the formal connection between RoPE-based attention and convolution to highlight its underlying inductive bias. We show that when the softmax operation is omitted, the attention becomes mathematically equivalent to a dynamic, content-aware convolution. Moreover, in this convolutional setting, the attention operator's complexity scales linearly with the number of tokens, akin to methods like Performer (Choromanski et al., 2020). This result reframes RoPE-attention as a mechanism that explicitly learns and applies dynamic, content-aware geometric filters.

## 2. Background: Transformers with Position Embeddings

The core of a Transformer is its self-attention mechanism, which computes outputs for a sequence of input features $\{f_i \in \mathbb{R}^C\}$ based on pairwise interactions. To perform spatial tasks, this operation must incorporate the position $p_i \in \mathbb{R}^n$ associated with each feature $f_i$. This positional information, often added via absolute or relative encodings, allows the model to learn relationships that respect geometric symmetries.

### 2.1. Vanilla Attention and Absolute Positioning

Given a sequence of input features $\mathbf{f}_i \in \mathbb{R}^C$, the self-attention layer first computes query, key, and value vectors via linear projections: $\mathbf{q}_i = \mathbf{W}^Q \mathbf{f}_i$, $\mathbf{k}_j = \mathbf{W}^K \mathbf{f}_j$,

$\mathbf{v}_j = \mathbf{W}^V \mathbf{f}_j$. Here, the learnable weight matrices are $\mathbf{W}^Q, \mathbf{W}^K \in \mathbb{R}^{C \times d}$ and $\mathbf{W}^V \in \mathbb{R}^{C \times C'}$. The output for the $i$-th feature, $\mathbf{y}_i \in \mathbb{R}^{C'}$, is a weighted sum of the value vectors, with weights determined by softmax-normalized dot products of queries and keys:

$$\mathbf{y}_i = \sum_{j=1}^{N} \text{attn}(\mathbf{q}_i, \mathbf{k}_j) \mathbf{v}_j$$

where

$$\text{attn}(\mathbf{q}_i, \mathbf{k}_j) = \text{softmax}_j \left( \mathbf{q}_i^\top \mathbf{k}_j \right) .$$

As this operation is permutation-equivariant, it is insensitive to the order of the inputs and must be modified to incorporate positional information for spatial tasks. A common approach is to use Absolute Positional Embeddings (APE), where a unique vector $\mathbf{E}(\mathbf{p}_i)$ is added to each input feature, $\mathbf{f}_i' = \mathbf{f}_i + \mathbf{E}(\mathbf{p}_i)$, before the linear projections are applied. The attention score is then computed from these position-aware features. However, since this interaction depends on absolute coordinates rather than relative positions, APE is not translation-equivariant.

### 2.2. Rotary Position Embeddings (RoPE)

RoPE achieves a more structured approach to position encoding (Su et al., 2024). Instead of adding a positional vector, RoPE modifies the query and key vectors with a position-dependent transformation, making the attention score explicitly dependent on relative positions.

This transformation is constructed by stacking 2D rotation matrices, giving RoPE its name. To apply RoPE with positions $\mathbf{p}$ in dimension $n > 1$, we use a set of $n$-dimensional frequency vectors $\Omega = \{\omega_k\}_{k=1}^{d/2}$, each defining a direction used to project $\mathbf{p}$ to 1D and a frequency used to apply 1D-RoPE in this direction. We obtain $d/2$ blocks,

$$\rho_{\omega_k}(\mathbf{p}) = \begin{pmatrix} \cos(\omega_k^\top \mathbf{p}) & -\sin(\omega_k^\top \mathbf{p}) \\ \sin(\omega_k^\top \mathbf{p}) & \cos(\omega_k^\top \mathbf{p}) \end{pmatrix}, \qquad (1)$$

which are stacked in a block-diagonal manner to form a single transformation matrix, $\rho_\Omega(\mathbf{p})$:

$$\rho_\Omega(\mathbf{p}) = \text{diag}(\rho_{\omega_1}(\mathbf{p}), \dots, \rho_{\omega_{d/2}}(\mathbf{p})) . \qquad (2)$$

Note that while $\rho_\Omega(\mathbf{p})$ is a high-dimensional rotation, this rotation is not related to rotations of the position $\mathbf{p}$. In fact, $\rho_\Omega$ is instead connected to translations of $\mathbf{p}$, formally discussed in Appendix A.

For a query $\mathbf{q}_i$ at position $\mathbf{p}_i$ and a key $\mathbf{k}_j$ at position $\mathbf{p}_j$, $\rho_\Omega$ is applied before the dot product. As the operator $\rho_\Omega$ is orthogonal and satisfies the homomorphism property[1] for

---

[1] The operator $\rho_\Omega(\mathbf{p})$ being orthogonal means its inverse is its transpose: $\rho_\Omega(\mathbf{p})^{-1} = \rho_\Omega(\mathbf{p})^\top$. The homomorphism property for the translation group satisfies $\rho_\Omega(\mathbf{p_i} + \mathbf{p_j}) = \rho_\Omega(\mathbf{p_i})\rho_\Omega(\mathbf{p_j})$.

translations, the interaction simplifies to depend only on relative positions:

$$(\rho_\Omega(\mathbf{p}_i)\mathbf{q}_i)^\top (\rho_\Omega(\mathbf{p}_j)\mathbf{k}_j) = \mathbf{q}_i^\top \rho_\Omega(\mathbf{p}_i)^\top \rho_\Omega(\mathbf{p}_j)\mathbf{k}_j \quad (3)$$
$$= \mathbf{q}_i^\top \rho_\Omega(\mathbf{p}_j - \mathbf{p}_i)\mathbf{k}_j \,.$$

This final form reveals the core property of RoPE. Although widely adopted for its empirical success, the mechanism's effectiveness is not coincidental; it directly embeds translation equivariance into the attention mechanism by making the score a function of content and relative positions. This powerful geometric inductive bias, often hidden within the standard Transformer framework, provides a principled reason for RoPE's strong performance (Chen et al., 2023; Dai et al., 2019). The formal construction of this operator from the first principles of group theory is detailed in Appendix A.

## 3. The Platonic Transformer

We generalize the principle of RoPE to obtain equivariance not only under continuous translations, but also discrete roto-reflections. We obtain roto-reflection equivariance by redefining the positional encoding relative to a set of reference frames defined as elements in a discrete subgroup $\mathcal{G} \subset O(n)$. Traditional RoPE-attention operates on a single global reference frame. Instead, we perform attention on multiple frames in parallel. A key advantage of our method is that it leaves the rope-attention mechanism and the overall computation graph unchanged from the traditional transformer.

### 3.1. Features relative to reference frames

Throughout the architecture, features are represented and processed *relative to the reference frames* defined by the elements of a discrete group $\mathcal{G} \subset O(n)$. Since input features are typically defined in a global frame of reference, they must first be *lifted* to become functions on the group $\mathcal{G}$. Specifically, each feature becomes a map $\mathbf{f}_i(\cdot) : \mathcal{G} \to \mathbb{R}^C$, where $\mathbf{f}_i(R)$ is the feature vector at point $i$ viewed from frame $R \in \mathcal{G}$. For the finite groups we consider, this map is represented as a tensor of shape $[|\mathcal{G}|, C]$. We denote this tensor simply as a flattened vector $\mathbf{f}_i \in \mathbb{R}^{|G|\cdot C}$ and use the functional notation $\mathbf{f}_i(\cdot)$ to emphasize its role as a feature map. As we will see, the flattened vector viewpoint is key to preserving the standard Transformer computation graph.

The lifting process depends on the geometric type of the input feature. Scalar features, being invariant to viewpoint, are lifted to constant functions by copying them across all frames. Vector features, in contrast, are expressed relative to each frame; for example, a single 3D vector feature $\mathbf{u} \in \mathbb{R}^3$ is lifted to a three-channel signal on the group via the transformation $\mathbf{f}(R) = R^{-1}\mathbf{u}$. All such lifted components can be concatenated, after which they are processed by the

subsequent equivariant, frame-dependent attention layers.

### 3.2. Weight-sharing across RoPE Embeddings

The key step for achieving equivariance to $\mathcal{G}$ as well as translations is making the RoPE operator itself dependent on the reference frames. This is achieved by projecting the position $\mathbf{p}_i$ of each input token $i$ onto $R$, which yields views $\mathbf{p}_i(R) = R^{-1}\mathbf{p}_i$ relative to each frame. As the queries $\mathbf{q}_i$, keys $\mathbf{k}_j$, and values $\mathbf{v}_j$ are obtained by applying equivariant linear projections (cf. Section 3.3) to the feature maps $\mathbf{f}_i$, they are also functions on the group. We can then compute the unnormalized attention scores from the perspective of frame $R$, which we denote as $s_{ij}(R)$:

$$s_{ij}(R) = \mathbf{q}_i(R)^\top \rho_\Omega(\mathbf{p}_j(R) - \mathbf{p}_i(R))\mathbf{k}_j(R) \quad (4)$$
$$= \mathbf{q}_i(R)^\top \rho_\Omega((\mathbf{p}_j - \mathbf{p}_i)(R))\mathbf{k}_j(R)\,. \quad (5)$$

Scores for each frame are computed *in parallel* as their own independent attention head. Note that we can also obtain $s_{ij}(R)$ by steering the base set of frequencies $\Omega$ instead of the positions $\mathbf{p}_i$, which we show in Appendix D. However, from our current perspective, the RoPE-attention mechanism itself remains completely unchanged from its traditional formulation in Eq. 3; only the relative positions $\mathbf{p}_i - \mathbf{p}_j$ are now defined relative to each reference frame $R$. The attention coefficients are obtained by applying the softmax to the scores $s_{ij}(R)$. The output $\mathbf{y}_i(R)$ for each token $i$ is then given as,

$$\mathbf{y}_i(R) = \sum_{j=1}^{N} \text{attn}_{ij}(R)\mathbf{v}_j(R)\,, \quad (6)$$
$$\text{where} \quad \text{attn}_{ij}(R) = \underset{j}{\text{softmax}}(s_{ij}(R))\,.$$

This process naturally results in an output tensor $\mathbf{y}_i \in \mathbb{R}^{|\mathcal{G}|\cdot C}$, where the features are defined relative to each frame. Notably, the base frequencies $\Omega$ of RoPE are shared across frames and this leads to the operator being equivariant to the roto-reflections in $G$, as we detailed in Appendix B.

### 3.3. Equivariant Linear Layers and Fixed Computational Graph

All linear transformations, including the query, key, and value projections ($\mathbf{W}^Q, \mathbf{W}^K, \mathbf{W}^V$), and any MLP blocks, must be equivariant. As our features can be viewed either as functions on the group, $\mathbf{f}_i(\cdot)$, or as flattened vectors, $\mathbf{f}_i \in \mathbb{R}^{|\mathcal{G}|\cdot C}$, we can describe the action of an equivariant linear layer $\Phi$ from both perspectives. From the flattened vector viewpoint, the layer is a standard matrix-vector multiplication, $\mathbf{y}_i = \mathbf{W}\mathbf{f}_i$. However, for this transformation to be equivariant, the weight matrix $\mathbf{W}$ cannot be arbitrary; it must have a specific, constrained structure.

The equivariance constraint is defined from the functional

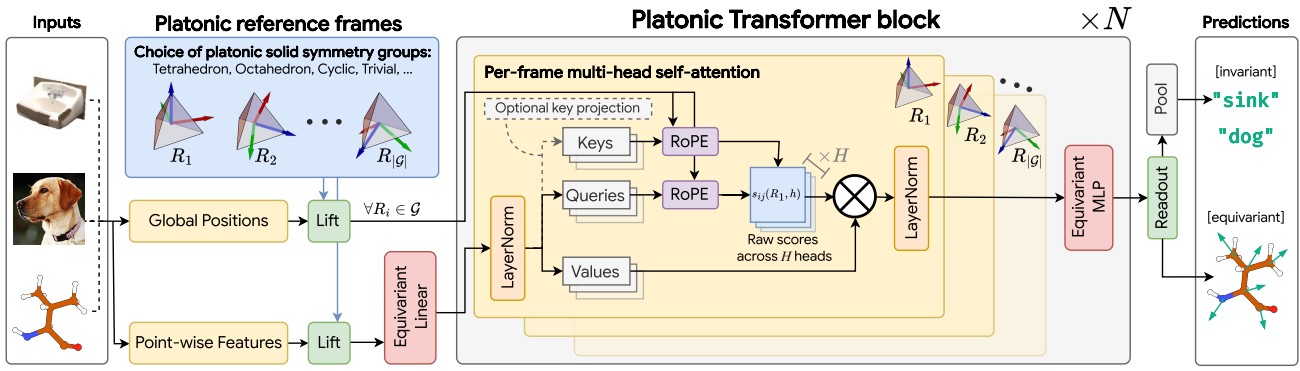

Figure 1. Visualization of Weight-Shared RoPE within the $N$-layer Platonic Transformer. Scalar and vector inputs are lifted to become functions on the platonic solid symmetry group of choice (here, the Tetrahedral group). The same multi-head self-attention mechanism is applied in parallel, with each instance rotating the features according to a different reference frame $R_i \in \mathcal{G}$. Choosing the trivial group as $\mathcal{G}$ reduces this framework to a standard Transformer.

viewpoint: for any group element $R \in \mathcal{G}$, the transformation must satisfy $\Phi(L_R \mathbf{f}_i) = L_R(\Phi(\mathbf{f}_i))$, where $L_R$ is the action of rotating the reference frames, i.e., $(L_R \mathbf{f}_i)(\tilde{R}) = \mathbf{f}_i(R^{-1}\tilde{R})$. This constraint is satisfied *if and only if* the layer's operation is a *group convolution* (Cohen et al., 2019, Thm. 3.1). This gives the layer a dual identity: it is a convolution over the group axis, which is mathematically equivalent to a matrix-vector multiplication with a structured, weight-shared matrix:

$$(\Phi(\mathbf{f}_i))(R) := \sum_{\tilde{R} \in \mathcal{G}} \mathbf{W}_{\text{group}}(R^{-1}\tilde{R})\,\mathbf{f}_i(\tilde{R}) \qquad (7)$$

$$\Longleftrightarrow \qquad \Phi(\mathbf{f}_i) := \mathbf{W}\mathbf{f}_i \qquad (8)$$

Here, $\mathbf{W}_{\text{group}} : \mathcal{G} \to \mathbb{R}^{C' \times C}$ is a learnable kernel defined on the group. The large matrix $\mathbf{W} \in \mathbb{R}^{(|\mathcal{G}| \cdot C') \times (|\mathcal{G}| \cdot C)}$ is a block matrix whose blocks are determined by the kernel values: $[\mathbf{W}]_{R,\tilde{R}} = \mathbf{W}_{\text{group}}(R^{-1}\tilde{R})$. This structure imposes a weight-sharing scheme where the interaction between input and output frames depends only on their *relative pose*, $R^{-1}\tilde{R}$. The layer is thus constrained to learn patterns from the geometric arrangement of features, rather than their absolute pose.

While the group convolution formulation makes the geometric inductive bias explicit, the matrix-vector viewpoint clarifies that this is in essence a principled *weight-sharing scheme* that preserves the computation graph of a standard linear layer from $|\mathcal{G}| \cdot C$ to $|\mathcal{G}| \cdot C'$ channels (we're still doing matrix-vector multiplication). A favorable side-effect, however, is that this structure reduces the parameter count from the $(|\mathcal{G}| \cdot C') \times (|\mathcal{G}| \cdot C)$ of an unconstrained layer to just $|\mathcal{G}| \cdot C' \cdot C$—a reduction by a factor of $|\mathcal{G}|$.

Crucially, by choosing the number of channels $C$ such that the effective feature dimension $C \cdot |\mathcal{G}|$ is held constant, the overall matrix dimensions are identical regardless of the

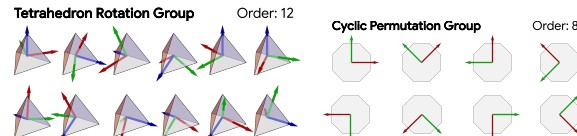

Figure 2. Elements of the symmetry groups of platonic solids form a subgroup of $SO(3)$.

group size. The trivial group $\mathcal{G} = \{\mathbf{e}\}$ illustrates the base case, where the operation collapses to a standard linear layer with a weight matrix $\mathbf{W}_{\text{group}}(\mathbf{e})$ of size $C' \times C$. The geometric inductive bias is therefore not introduced by adding new, complex modules, but by imposing a structure on the weights of existing ones.[2]

With all components of the architecture now defined as equivariant operations, we can formally state the key property of the full model, namely equivariance under the discrete group $\mathcal{G} \subset O(n)$.

**Proposition 3.1** (End-to-End Equivariance). *Our proposed Transformer architecture is an equivariant model. A global roto-reflection $R \in \mathcal{G}$ applied to the input point cloud results in a corresponding transformation $L_R$ of the final output feature maps.*

The proof is given in Appendix B.

### 3.4. Frame Selection Via Platonic Solids

The final step is to select a suitable subgroup $\mathcal{G} \subset O(n)$ to serve as the reference frames. We select them from the dis-

---

[2]This structure can even give computational benefits, by implementing the linear layers in the Fourier domain of $\mathcal{G}$ (Bökman et al., 2025). In Appendix O, we find that at marginally higher channel counts than used in this paper, a Fourier implementation leads to greatly improved training throughput, indicating that this is a promising direction for future research.

crete symmetry groups of regular polygons and polyhedra, with different considerations for 2D and 3D as illustrated in Figure 2.

*In 3D*, we restrict our frames to the finite *rotational* symmetry groups ($\mathcal{G} \subset SO(3)$) of the Platonic solids: the *tetrahedral* (12 rotations), *octahedral* (24 rotations), and *icosahedral* (60 rotations) groups. While these solids have larger full symmetry groups that include reflections (e.g., 24 total symmetries for the tetrahedron), we focus on the purely rotational subgroups for a more tractable structure.

*In 2D*, we consider discrete subgroups of $O(2)$, which correspond to the symmetries of regular polygons. This includes both the rotation-only *cyclic groups* ($C_n$) and the *dihedral groups* ($D_n$), which contain both rotations and reflections. Here $n$ denotes the group's order. This discrete subgroup approach is advantageous for two reasons. First, it provides a finite set of frames that forms a structured and approximately uniform discretization of the underlying continuous spaces of orientations ($SO(3)$ in 3D and $O(2)$ in 2D). Second, and more critically, these frames form a group. This is essential for maintaining a meaningful geometric structure, as it ensures that layers can operate equivariantly, keeping features coherently defined relative to our chosen frames throughout the network.

The advantage of working with a *finite* group $\mathcal{G}$ is that its operations can be handled discretely and efficiently using *Cayley tables*. We assign a unique index $i \in \{0, \dots, |\mathcal{G}|-1\}$ to each rotation $R_i \in \mathcal{G}$. The group product $R_i R_j = R_k$ can then be precomputed and stored in the Cayley table, a simple look-up table where $\text{Cayley}[i, j] = k$. This discrete formalism makes the group action on our feature maps, which are functions on the group $f : \mathcal{G} \to \mathbb{R}^C$, extremely efficient. A rotation of this feature map by an element $R_i$, defined by the action $(L_{R_i} f)(R_j) = f(R_i^{-1} R_j)$, simplifies to a permutation of the feature tensor's entries. With the Cayley table, the new feature at position $j$ is simply copied from the old feature at position $k = \text{Cayley}[\text{inverse}[i], j]$.

## 4. Inductive Bias of Platonic Transformers

This section examines the Platonic Transformer's structural inductive biases. We highlight its interpretation as a dynamic group convolution and its equivariant attention, contrasting these with approaches based on invariant attention.

### 4.1. Platonic Transformer as Dynamic Group Convolution

The use of RoPE in a linear attention setting establishes a deep connection to convolution. Specifically, the mechanism implements an adaptive convolution where the kernel is synthesized on-the-fly. This dynamic kernel is expressed as an expansion in a sparse Fourier basis, defined by the

RoPE frequencies, and the coefficients for this basis expansion are provided by the query vectors. This makes the convolution content-aware. We formalize this as follows (proof in Appendix C.1).

**Proposition 4.1** (Linear RoPE Attention as Dynamic Convolution)**.** *Consider a standard linear attention layer using RoPE with constant key vectors ($\mathbf{k}_j = \mathbf{1}$). The layer's output $\mathbf{y}_i$ is mathematically equivalent to a dynamic convolution:*

$$\mathbf{y}_i = \sum_{j=1}^{N} \phi_{\mathbf{q}_i}(\mathbf{p}_j - \mathbf{p}_i)\mathbf{v}_j\,, \tag{9}$$

*where the dynamic kernel $\phi_{\mathbf{q}_i}$ is given by the inverse sparse Fourier transform:*

$$\phi_{\mathbf{q}_i}(\Delta\mathbf{p}) = \sum_{k=1}^{d/2} \left[ a_k(\mathbf{q}_i) \cos(\omega_k^\top \Delta\mathbf{p}) + b_k(\mathbf{q}_i) \sin(\omega_k^\top \Delta\mathbf{p}) \right]. \tag{10}$$

*The Fourier coefficients are given by the linear projections $a_k(\mathbf{q}_i) = q_{i,2k-1} + q_{i,2k}$ and $b_k(\mathbf{q}_i) = q_{i,2k} - q_{i,2k-1}$, where $q_{i,m}$ is the $m$-th element of the query vector $\mathbf{q}_i$.*

*Remark* 4.2 (Purely Geometric vs. Mixed Kernels)*.* This result recasts the query's role: rather than simply probing for content, $\mathbf{q}_i$ enables the parameters to construct a unique geometric filter. The formulation of the key vector is a design choice. The constant-key formulation ($\mathbf{k}_j = \mathbf{1}$) forces the model to learn purely geometric, content-adaptive convolution operators. In contrast, a learned key ($\mathbf{k}_j = \mathbf{W}^K \mathbf{f}_j$) results in a mixed kernel whose coefficients depend on both query and key features, and thus entangles geometry and signal, possibly increasing expressivity while making score magnitudes and optimization more sensitive unless stabilized.

This gives a practical expressivity–stability trade-off. Fixed keys enforce a purely geometric, content-adaptive kernel and were the most robust choice in our QM9 ablations (Appendix M). We hypothesize that this robustness is especially useful for molecular tasks, where the target is governed by universal *physical principles* that are largely functions of geometry and atom types, whereas computer vision tasks such as ScanObjectNN often involve learning *statistical correlations* between local appearance and global shape. A mixed kernel from learned keys can entangle these physical principles with instance-specific chemical environments, creating a more delicate optimization problem as the model attempts to learn a general physical law while simultaneously fitting local molecular context. In computer vision, this same entanglement can be beneficial, as learning the statistical interplay between features and geometry is often the primary objective. We therefore use fixed keys for the smaller QM9 setting, while for the larger OMol25 setting we use learned keys together with QK normalization, which controls query/key magnitudes before the dot product and

stabilizes training. Regardless, the convolution perspective further leads to a key practical advantage.

**Corollary 4.3** (Linear-Time Complexity). *The dynamic convolution in Proposition 4.1 can be computed in $O(N)$ time, where $N$ is the number of tokens or points in the point cloud. This offers a scalable alternative to standard attention, which has a quadratic complexity of $O(N^2)$.*

Within our Platonic Transformer, this entire mechanism is lifted to operate over the reference frames defined by a group $\mathcal{G}$. Consequently, the operator becomes an adaptive *group convolution* (proof in Appendix C.3), where the kernel is steered by the group elements/reference frames.

### 4.2. Invariant vs. Equivariant Attention Score

Our approach implements an *equivariant attention* mechanism, where the attention pattern is orientation-dependent. This contrasts with methods using an *invariant attention* score, which applies the same pattern from all orientations (Fuchs et al., 2020; Chen & Villar, 2022; Assaad et al., 2023; Frank et al., 2024; Knigge et al., 2024; Kundu & Kondor, 2025; Nordström et al., 2025).

For multi-head attention with $H$ heads, let $\mathbf{q}_i(R, h)$, $\mathbf{k}_j(R, h)$, and $\mathbf{v}_j(R, h)$ denote the projected query, key, and value vectors for head $h$ from the perspective of frame $R$. In our equivariant approach, the raw scores $s_{ij}(R, h)$ are passed directly to the softmax. This allows the model to learn orientation-dependent attention patterns, making it a more expressive formulation that retains the rich geometric information in the features. The output is an equivariant feature map on the group:

$$\mathbf{y}_i(R, h) = \sum_{j=1}^{N} \underset{j}{\text{softmax}} \big( \underbrace{s_{ij}(R, h)}_{R-\text{dependent}} \big) \mathbf{v}_j(R, h), \qquad (11)$$

$$s_{ij}(R, h) = \mathbf{q}_i(R, h)^\top \rho_{\Omega_h}((\mathbf{p}_j - \mathbf{p}_i)(R)) \mathbf{k}_j(R, h). \quad (12)$$

In practice, this is efficiently implemented by treating the $|\mathcal{G}|$ perspectives as an independent set of attention heads. Tensors are reshaped so that the group and head dimensions are merged, e.g., to a shape of $[B, N, |\mathcal{G}| \cdot H, C_h]$, before the dot product calculation.

In an invariant attention score, a single attention pattern is created by pooling the raw scores over the group axis before the softmax, akin to the *symmetrization* in the RoPE-based approach of Frank et al. (2024). These invariant attention scores are then applied to the original equivariant value vectors. The resulting output is still equivariant, but it is derived from an *orientation-agnostic attention pattern*:

$$\mathbf{y}_i(R, h) = \sum_{j=1}^{N} \underset{j}{\text{softmax}} \big( \underbrace{s_{ij}^{\text{inv}}(h)}_{R-\text{agnostic}} \big) \mathbf{v}_j(R, h), \qquad (13)$$

$$\text{where} \quad s_{ij}^{\text{inv}}(h) = \sum_{R \in \mathcal{G}} s_{ij}(R, h).$$

Although simpler, this formulation sacrifices the model's ability to attend to features in an orientation-dependent manner. Implementing Eq. 13 can be done by reshaping tensors so that the group and channel dimensions are merged, to shape $[B, N, H, |\mathcal{G}| \cdot C_h]$, as then the dot-product in $s_{ij}$ and the sum in $s_{ij}^{\text{inv}}$ are simultaneously computed when taking the dot-product between queries and keys. For a fully invariant output, one could additionally average the value vectors $\mathbf{v}_j(R, h)$ over the group to further collapse the geometric representation.

## 5. Experiments

To validate our proposed architecture, we conduct a series of experiments across a number of different tasks and datasets. Our evaluation is structured to analyze the role of the equivariance inductive bias by categorizing tasks into two distinct settings based on their inherent geometric properties.

First, for tasks with inherent symmetry, such as those in QM9 (Ramakrishnan et al., 2014) and OMol25 (Levine et al., 2025), the underlying molecular systems have no canonical orientation. Their properties are determined by the relative positions of atoms and are independent of the global coordinate system. Since the physical laws governing these molecular properties are E(3)-symmetric, equivariance becomes a fundamental requirement for a model to generalize efficiently (Fuchs et al., 2020; Bronstein et al., 2021; Batzner et al., 2022; Pacini et al., 2025; Vadgama et al., 2025). We refer to this category as *Equivariant Tasks*.

Second, for tasks involving datasets with a canonical orientation, like CIFAR-10 (Krizhevsky, 2009) and ScanObjectNN (Uy et al., 2019), strict end-to-end equivariance is not required (the images/objects are aligned w.r.t. a canonical up-direction). These problems nevertheless provide a testbed to investigate if the geometric inductive bias of our model, enforced by weight-sharing, improves performance on its own merits. We refer to these as *Non-Equivariant Tasks*. We provide additional results on ImageNet-1K (Deng et al., 2009) in Appendix G.

### 5.1. Experimental Setup

All Platonic Transformer variants are built upon RoPE, making them inherently *translation-equivariant*. The degree of rotational equivariance is then determined by the choice of a discrete symmetry group $\mathcal{G} \subset O(n)$ that defines the set of reference frames. For instance, selecting the trivial group ($\mathcal{G} = \{\mathbf{e}\}$) results in a purely translation-equivariant model ($T(n)$); it uses only the identity frame. Choosing the rotational symmetry group of the Tetrahedron provides 12 reference frames, making the model approximately $SE(n)$-equivariant, or $E(n)$-equivariant when including reflections too.

*Table 1.* CIFAR-10 Accuracy (%).

| Group | Attention Acc. (↑) | Conv Acc. (↑) | # Params |
|---|---|---|---|
| $\{\mathbf{e}\}^*$ | $91.65_{\pm 0.27}$ | $86.65_{\pm 0.24}$ | $85.1M$ |
| $C_4$ | $92.30_{\pm 0.14}$ | $87.39_{\pm 0.60}$ | $21.3M$ |
| $C_6$ | $92.66_{\pm 0.04}$ | $\mathbf{87.62}_{\pm 0.26}$ | $14.2M$ |
| $D_2$ | $92.05_{\pm 0.09}$ | $86.11_{\pm 0.53}$ | $21.3M$ |
| $D_3$ | $\mathbf{92.78}_{\pm 0.28}$ | $87.47_{\pm 0.07}$ | $14.2M$ |
| Flop | $92.35_{\pm 0.16}$ | $86.70_{\pm 0.51}$ | $42.6M$ |

*Table 2.* ScanObjectNN Overall Acc. (%).

| Group | Attention Acc. (↑) | Conv Acc. (↑) |
|---|---|---|
| $\{\mathbf{e}\}^*$ | $83.68_{\pm 0.47}$ | $77.46_{\pm 0.51}$ |
| Flop | $84.18_{\pm 0.34}$ | $78.74_{\pm 0.54}$ |
| Tetrahedron | $\mathbf{86.11}_{\pm 0.28}$ | $\mathbf{79.00}_{\pm 0.59}$ |

*$^*$Platonic Transf. w. group $\{\mathbf{e}\}$ is a ViT w. RoPE*

For fair comparison, we match the computational cost between $SE(n)$ and $T(n)$ models by equating our group-based parallelism with standard multi-head attention. For instance, an $SE(n)$ model using the 12-element tetrahedral group with one head per frame is benchmarked against a $T(n)$ baseline with 12 total heads (details in App. F). For certain tasks, symmetries can be conditionally broken by using APE or providing an *external reference frame*, yet internal layers critically retain principled weight-sharing. Using external frames to break symmetry and APE to provide geometric information are effective strategies, allowing a model to benefit from geometric inputs without being end-to-end constrained by full equivariance (Vadgama et al., 2025).

For CIFAR-10 and ScanObjectNN, we conducted a comprehensive sweep to find the optimal configuration. In contrast, for OMol25, we used a sequential process: first, we identified the best architecture via an extensive sweep on property prediction on QM9, then transferred these hyperparameters to OMol25 for further refinement with a one-million subset before full training (see Appendix I-K).

We emphasize that the primary objective of these experiments is not the pursuit of state-of-the-art (SOTA) performance through exhaustive architectural engineering. Rather, we aim to empirically validate that our proposed lifting mechanism can convert a standard Vision Transformer into its equivariant counterpart while maintaining a strictly identical computational budget. Through this controlled comparison, we demonstrate that Platonic Transformers consistently achieve superior or competitive performance relative to task-specific SOTA methods, thereby establishing the practical efficacy of principled geometric weight-sharing.

### 5.2. Non-Equivariant Tasks

**CIFAR-10** The results of our ablation study on CIFAR-10 are presented in Table 1. Flop denotes the group of left-to-right flips (Bökman et al., 2025). The findings indicate that incorporating 2D rotational symmetries provides a tangible benefit over the translation-only baseline (the $\mathcal{G} = \{\mathbf{e}\}$ model, which is equivalent to a standard Vision Transformer). This suggests that even for general-purpose vision tasks without an end-to-end equivariance requirement,

equivariance proves to be an important inductive bias. This may be explained by the fact that even though images have a canonical pose (e.g. with the sky at the top), equivariance allows for internal weight-sharing and thus the reuse of patterns (edges, parts, objects) that may appear at arbitrary orientations within an image. More concretely, the strongest-performing configurations in both regimes are also the most parameter-efficient ones: $D_3$ achieves the best attention accuracy, while $C_6$ achieves the best convolutional accuracy, and both use only $14.2M$ parameters compared to $85.1M$ for the $\mathcal{G} = \{\mathbf{e}\}$ baseline. This supports the view that the gain is not merely due to increased capacity, but rather to the symmetry-induced weight sharing itself. Interestingly, the largest tested symmetry groups provide the best results on this non-equivariant task, suggesting that stronger geometric tying can act as an effective regularizer even when the final prediction is not required to be equivariant. Finally, the comparison between the full attention and linear-convolutional variants shows a significant impact of attention over the linear-complexity dynamic convolution counterpart. Averaged over all groups in Table 1, attention improves accuracy by $5.31$ percentage points over the convolutional variant, in which the softmax is omitted (cf. Prop. 4.1).

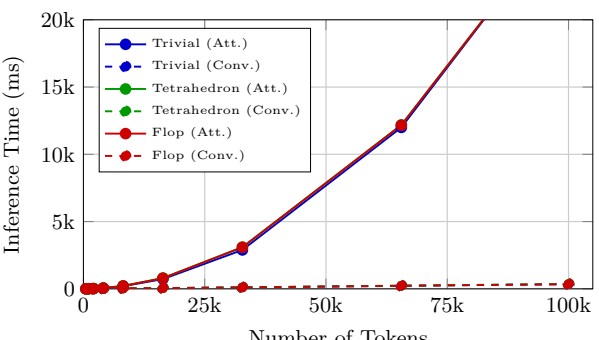

*Figure 3.* In convolutional mode, the Platonic Transformer scales linearly with sequence length matching its attention mode.

**ScanObjectNN** On the ScanObjectNN point cloud classification task, we test the effectiveness of 3D symmetry groups ($\{\mathbf{e}\}$, Flop, and Tetrahedron) in a realistic setting with occlusions and significant orientation variability. Similar to CIFAR-10, this is not a strictly equivariant task, yet the results in Table 2 again highlight the impact of equivariance and weight sharing. Tetrahedron achieves the best

*Table 3.* PoseBusters pass rates (%) on QM9 molecule generation.

| Test | Symphony | Eq. Diff. | ADiT | ZATOM-1 | ZATOM-1-WD | DiP-{e} | DiP-Tetra. |
|------|----------|-----------|------|---------|-----------|---------|-----------|
| Atoms connected | 99.92 | 99.88 | 99.70 | 99.98 | **100** | **100** | **100** |
| Bond angles | 99.56 | **99.98** | 99.85 | 99.95 | 99.91 | 99.87 | 99.91 |
| Bond lengths | 98.72 | **100** | 99.41 | 99.97 | 99.94 | 99.92 | 99.95 |
| Aromatic ring flat | **100** | 100 | 100 | 100 | 100 | 100 | 100 |
| Double bond flat | 99.07 | 98.58 | 99.98 | 99.99 | **100** | **100** | **100** |
| Internal energy | 95.65 | 94.88 | 95.86 | 99.78 | 99.79 | 99.87 | **99.89** |
| No steric clash | 98.16 | 99.79 | 99.79 | 99.81 | 99.84 | **100** | **100** |

performance for both attention and convolution, improving over the {e} baseline by 2.43 and 1.54 percent, respectively. This again suggests that stronger symmetry-induced weight sharing is beneficial even when the final classification task does not require end-to-end equivariance. Consistent with the CIFAR-10 results, we again observe the superiority of attention over its linear-convolutional counterpart, with an average improvement of 6.26 percentage points across all groups. Nevertheless, the linear-time convolutional variant provides a significant speed-up, which can be critical for efficiently processing large point clouds. This demonstrates the versatility of our approach in adapting to different computational and modeling requirements in 3D computer vision, as demonstrated in Figure 3. Also note that the computational cost is independent of the chosen symmetry group.

*Table 4.* Molecule generation results on QM9.

| Group | Validity % (↑) | Unique % (↑) | Mol. Stab. (↑) | Atom Stab. (↑) |
|-------|----------------|--------------|----------------|----------------|
| DiP-{e} | 97.09 | 96.56 | 91.54 | 99.29 |
| DiP-Tetrahedron | **98.43** | 97.02 | **95.26** | **99.56** |
| **Reference Method** | | *results from cited works* | | |
| QM9-only ZATOM-1 [45] | 92.88 | 97.71 | – | - |
| Jointly trained ADiT [32] | 94.45 | 97.82 | – | - |
| QM9-only ADiT [32] | 92.19 | 97.90 | – | - |
| Symphony [20] | 83.50 | 97.98 | 83.50 | - |
| GeoLDM [71] | 93.80 | **98.82** | 89.4 | 98.9 |
| EDM [31] | 91.90 | 98.69 | 82.0 | 98.7 |

## 5.3. Equivariant Tasks

**Molecule generation on QM9.** We replace the standard Transformer in the DiffusionTransformer (DiT) (Peebles & Xie, 2023) with our Platonic Transformer, which we call DiP, and follow the training regimen from (Joshi et al., 2025). We operate at an all-atom resolution, including explicit Hydrogen atoms. We use evaluation procedure detailed in (Joshi et al., 2025) for RDKit-based validity and uniqueness, while molecule and atom stability are computed using the EDM validation pipeline (Hoogeboom et al., 2022).

We present results in Table 4. DiP is at least on par with jointly-trained and QM9-only variants of ADiT (Joshi et al., 2025) in terms of validity, with DiP-Tetrahedron achieving the highest validity among the compared methods. It also outperforms other baselines like GeoLDM (Xu et al., 2023), Equivariant Diffusion (EDM) (Hoogeboom et al., 2022), and Symphony (Daigavane et al., 2024) in validity, molecular stability, and atom stability.

In Table 3, we additionally report PoseBusters (Buttenschoen et al., 2023) sanity-check pass rates for QM9 molecule generation. The table reports pass rates on seven sanity checks for 10,000 sampled molecules, and all models explicitly generate hydrogen atoms. Both DiP variants achieve near-saturated pass rates across all checks, including perfect scores on atom connectivity, aromatic ring flatness, double-bond flatness, and steric-clash checks. This shows that the generated molecules are not only valid under standard metrics, but also geometrically plausible under stricter structural checks.

Our results present a nuanced view of equivariance in generative diffusion. While the trivial (DiP-{e}) and equivariant (DiP-Tetrahedron) models achieve close validity scores, the equivariant variant yields significantly higher molecular stability.This suggests that standard validity metrics may be overly lenient, whereas stability, which accounts for proximity, valency, and charge, offers a more rigorous measure of physical plausibility. The PoseBusters results provide a complementary sanity check, showing that these gains in stability do not come at the cost of basic 3D geometric consistency. Furthermore, the equivariant model exhibited superior training stability at large hidden dimensions and converged faster, consistent with the learning dynamics observed by Vadgama et al. (2025).

Notably, the validity gap between DiP-{e} and DiP-Tetrahedron is modest, whereas the convolution-based framework in (Vadgama et al., 2025) showed a pronounced 11.5% difference between non-equivariant and equivariant variants (We provide an ablation on the effects of symmetry group size in Appendix F.1.).This suggests Transformers may more effectively learn equivariant tasks from unconstrained parameterizations, particularly at scale. Since performance gaps often diminish with increased model size, a large-scale architecture may explain this similarity. While the Platonic Transformer demonstrates strong generative capabilities, further investigation is needed to characterize the molecular scaling laws governing its data and efficiency.

**Property prediction on QM9.** We evaluate on property prediction on QM9 in Appendix H, obtaining competitive results for this task as well.

*Table 5.* Performance on OMol25 4M structure-to-force/energy prediction. We report parameters, training throughput (atoms/sec), and validation MAE for forces, total energy, and energy per atom.

| Model | Epochs | Params | Throughput | Force | Energy | E/Atom |
|---|---|---|---|---|---|---|
| Tetrahedron | 20 | 60M | 33,878 | 11.15 | 83.02 | 1.41 |
| Octahedron | 20 | 30M | **34,023** | 12.28 | 94.61 | 1.62 |
| Octahedron | 20 | 60M | 18,121 | 10.51 | 82.57 | 1.42 |
| eSEN-sm[39] | 20 | 6M | 12,834 | 13.11 | 153.54 | 2.03 |
| Tetrahedron | 80 | 60M | 33,878 | **9.33** | **63.84** | **1.12** |
| eSEN-sm[39] | 80 | 6M | 12,834 | 12.64 | 107.78 | 1.66 |

*Table 6.* ProteinMD Force MSE ($\downarrow$).

| Group | Backbone level | Atom level |
|---|---|---|
| {e} | $2.11_{\pm 0.030}$ | $2.60_{\pm 0.019}$ |
| Tetrahedron | $1.80_{\pm 0.004}$ | $2.36_{\pm 0.011}$ |
| **Reference Method** | *results from Moskalev et al. [46]* | |
| EGNN [57]* | $2.25_{\pm 0.001}$ | $2.72_{\pm 0.003}$ |
| FastEGNN [74]* | $1.84_{\pm 0.002}$ | – |
| G-Transformer [46]* | $2.45_{\pm 0.037}$ | $3.67_{\pm 0.640}$ |
| G-Hyena [46]* | $1.80_{\pm 0.009}$ | $2.49_{\pm 0.037}$ |

**OMol25** To validate the scalability and performance of our proposed architecture, we evaluate our model with the best hyperparameters on the large-scale OMol25 dataset. We train on the OMol25 4M training set and report errors on the validation set. Table 5 compares model size and validation errors under 20 and 80-epoch settings using a cosine annealing learning rate schedule. Throughput is measured for forward and backward passes on real OMol dynamic batches. We report force MAE, total energy MAE, and energy-per-atom MAE.

The 20-epoch results show that both 60M-parameter Platonic Transformer variants outperform the eSEN-sm baseline in accuracy while also being substantially faster. In particular, the Tetrahedron model improves force MAE from 13.11 to 11.15 and energy MAE from 153.54 to 83.02, while achieving a $2.6\times$ higher throughput. Increasing the symmetry group from Tetrahedron to Octahedron reduces the parameter count from 60M to 30M through stronger weight sharing, while keeping throughput essentially unchanged. This comes with a small reduction in performance, which may be due to the reduced number of parameters rather than the larger symmetry group itself. To separate the effect of symmetry resolution from model capacity, we therefore also scale the Octahedron model back to 60M parameters. This improves force MAE to 10.51, the best 20-epoch force result in the table, although at the expected cost of lower throughput.

Following the common OMol25 evaluation setting, we also train the Tetrahedron model and eSEN-sm from scratch for 80 epochs. The eSEN-sm results are consistent with, and slightly stronger than, comparable reports under a similar training recipe in (Qu et al., 2026; Levine et al., 2025), which confirms that our implementation and optimization setup are reliable. Under the same setting, the Tetrahedron model substantially outperforms eSEN-sm across all error metrics, reducing force MAE from 12.64 to 9.33, energy MAE from 107.78 to 63.84, and energy-per-atom MAE from 1.66 to 1.12, while maintaining a $2.6\times$ higher throughput. Taken together, these results show that Platonic Transformers, despite simply being equivariance-constrained vanilla Transformers, form a scalable and efficient class of models for interatomic potentials, outperforming a state-of-the-art architecture specifically tailored to molecular force and energy prediction.

**ProteinMD** We evaluate our model on molecular dynamics dataset processed from MDAnalysis (Han et al., 2022).It models the equilibrium-time evolution of protein structures, where atom interactions depend on both local and long-range geometry. Data is processed with MDAnalysis from an AdK equilibrium MD trajectory (Seyler & Beckstein, 2017) yielding 4,186 protein structures with trajectories. We report results on backbone (855 atoms) and all-atom (3,341 atoms) variants. The results are presented in Table 6, where we see that we outperform the previous state-of-the-art G-Hyena (Moskalev et al., 2025) as well as their equivariant transformer baseline.

**Limitations** While we have evaluated Platonic Transformers on a broad range of experiments, the performance at extreme scale remains uncharacterized. Further, we only guarantee equivariance w.r.t. subgroups of $O(n)$, in specific applications full equivariance may be preferred.

# 6. Conclusion

We introduce the Platonic Transformer, a framework that achieves approximate $E(n)$ equivariance without compromising the flexibility and scalability of the standard Transformer architecture. By combining Rotary Position Embeddings (RoPE) with a new frame-dependent attention mechanism—where attention is computed relative to reference frames from Platonic solid symmetry groups—we integrate a powerful geometric inductive bias while preserving the original computation graph and cost. This approach demonstrates that principled equivariance and modern scalability are not mutually exclusive. Furthermore, our analysis reveals a formal equivalence to dynamic group convolution with linear complexity, enabling a highly scalable, linear-time variant for large-scale tasks. In many scientific domains, equivariance represents a "Platonic ideal" — an essential physical principle a model should respect. By eliminating the trade-off between this principled design and computational efficiency, the Platonic Transformer makes this ideal a practical and scalable reality.

## Impact Statement

The primary potential negative impacts are those common to advanced generative modeling and scientific ML, namely the risk of dual-use (e.g., repurposing molecular generation for harmful compounds) and the environmental cost of large-scale training on datasets like OMol25.

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

# Appendix Contents

# A. Rotary Position Embeddings from a Group Theoretical Perspective

A fundamental challenge in geometric deep learning is creating position representations that respect underlying symmetries. For data in $\mathbb{R}^d$, our goal is to define a high-dimensional position embedding, $\mathbf{E} : \mathbb{R}^d \to \mathbb{R}^{d'}$, that is equivariant to translations. This requires that for any translation vector $\mathbf{p}$, the embedding transforms predictably: $\mathbf{E}(\mathbf{p}_0 + \mathbf{p}) = \boldsymbol{\rho}(\mathbf{p})\mathbf{E}(\mathbf{p}_0)$, where $\boldsymbol{\rho}(\mathbf{p})$ is a linear transformation. Group representation theory provides the formal tools to construct such embeddings.

## A.1. The Theoretical Toolkit

To proceed, we first define the essential concepts required for our construction.

**Definition A.1** (Representation). A linear representation of a group $\mathcal{G}$ on a vector space $V$ is a group homomorphism $\rho : \mathcal{G} \to \mathrm{GL}(V)$, where $\mathrm{GL}(V)$ is the general linear group of invertible linear transformations on $V$.

To ensure that the positional encoding does not arbitrarily amplify or diminish feature magnitudes, which would destabilize learning, we require the representations to be length-preserving. This leads to the concept of a unitary representation.

**Definition A.2** (Unitary Representation). A representation $\rho$ is **unitary** if it maps group elements to unitary operators, i.e., $\rho : \mathcal{G} \to \mathrm{U}(V)$. For real-valued representations, this corresponds to orthogonality, $\rho(g)^{-1} = \rho(g)^\top$.

Just as a complex signal can be decomposed into pure frequencies, a general representation can be broken down into fundamental building blocks known as irreducible representations (irreps).

**Definition A.3** (Irreducible Representation). An **irreducible representation** (irrep) is a representation acting on a vector space $V$ that has no non-trivial invariant subspaces.

## A.2. Constructing the RoPE Operator

With these formal tools, we can now build the RoPE operator. The irreps of the translation group $(\mathbb{R}^d, +)$ are indexed by a frequency vector $\boldsymbol{\omega} \in \mathbb{R}^d$ and are given by one-dimensional, unitary representations:

$$\rho_{\boldsymbol{\omega}}(\mathbf{p}) = e^{i\boldsymbol{\omega}^\top \mathbf{p}}. \tag{14}$$

This exponential form is the unique continuous solution to the group's homomorphism property, $\rho(\mathbf{p}_1 + \mathbf{p}_2) = \rho(\mathbf{p}_1)\rho(\mathbf{p}_2)$, where the imaginary exponent ensures unitarity.

However, neural networks typically operate on real numbers. We can obtain a real-valued irrep by combining pairs of conjugate frequencies, $\boldsymbol{\omega}_k$ and $-\boldsymbol{\omega}_k$. This yields a 2D irreducible representation that takes the familiar form of a rotation matrix:

$$\rho_{\boldsymbol{\omega}_k}(\mathbf{p}) = \begin{pmatrix} \cos(\boldsymbol{\omega}_k^\top \mathbf{p}) & -\sin(\boldsymbol{\omega}_k^\top \mathbf{p}) \\ \sin(\boldsymbol{\omega}_k^\top \mathbf{p}) & \cos(\boldsymbol{\omega}_k^\top \mathbf{p}) \end{pmatrix}. \tag{15}$$

To create a high-dimensional embedding, we simply select a set of frequencies $\Omega = \{\boldsymbol{\omega}_k\}_{k=1}^{d'/2}$ and stack these 2D rotation blocks along the diagonal of a larger matrix. This results in a single, block-diagonal transformation that correctly and equivariantly updates the entire embedding for a given translation $\mathbf{p}$:

$$\boldsymbol{\rho}_\Omega(\mathbf{p}) = \mathrm{diag}(\rho^{\boldsymbol{\omega}_1}(\mathbf{p}), \ldots, \rho_{\boldsymbol{\omega}_{d'/2}}(\mathbf{p})). \tag{16}$$

This is the core mechanism behind Rotary Position Embeddings. Its structure guarantees both equivariance and computational efficiency, as each 2D component can be rotated independently.

**Definition A.4** (Rotary Position Embedding (RoPE) Operator). The RoPE operator $\boldsymbol{\rho}_\Omega(\mathbf{p})$ for a position $\mathbf{p} \in \mathbb{R}^d$ is the block-diagonal rotation matrix defined above (Equation 16), constructed from a set of frequencies $\Omega$. The application of RoPE to a feature vector $\mathbf{f} \in \mathbb{R}^{d'}$ is defined as the matrix-vector product: $\boldsymbol{\rho}_\Omega(\mathbf{p})\mathbf{f}$. For this operation to be well-defined, the feature dimension $d'$ must be even.

## A.3. Translation Invariance in Attention

While the RoPE operator provides an *equivariant* transformation for feature vectors, its crucial benefit within the Transformer architecture is that it makes the attention score *invariant* to global translations. This property ensures that the attention mechanism only considers the relative positions of tokens, which is the inductive bias we seek. We formalize this key result below.

**Proposition A.5** (Translation Invariance of the RoPE Attention Score). *The attention score computed using RoPE,* $\text{attn}(\mathbf{q}, \mathbf{k}, \Delta\mathbf{p}) = \mathbf{q}^\top \boldsymbol{\rho}_\Omega(\Delta\mathbf{p})\mathbf{k}$, *where* $\Delta\mathbf{p} = \mathbf{p}_j - \mathbf{p}_i$, *is invariant to a global translation of the coordinate system.*

*Proof.* Let the positions $\mathbf{p}_i$ and $\mathbf{p}_j$ be translated by an arbitrary vector $\mathbf{t} \in \mathbb{R}^d$, resulting in new positions $\mathbf{p}'_i = \mathbf{p}_i + \mathbf{t}$ and $\mathbf{p}'_j = \mathbf{p}_j + \mathbf{t}$. The new relative displacement vector, $\Delta\mathbf{p}'$, is:

$$\Delta\mathbf{p}' = \mathbf{p}'_j - \mathbf{p}'_i = (\mathbf{p}_j + \mathbf{t}) - (\mathbf{p}_i + \mathbf{t}) = \mathbf{p}_j - \mathbf{p}_i = \Delta\mathbf{p}. \tag{17}$$

Since the relative displacement vector is unchanged by the global translation, the RoPE operator applied to it also remains unchanged: $\boldsymbol{\rho}_\Omega(\Delta\mathbf{p}') = \boldsymbol{\rho}_\Omega(\Delta\mathbf{p})$. Consequently, the attention score, which depends only on the content vectors and this operator, is invariant to the translation:

$$\mathbf{q}^\top \boldsymbol{\rho}_\Omega(\Delta\mathbf{p}')\mathbf{k} = \mathbf{q}^\top \boldsymbol{\rho}_\Omega(\Delta\mathbf{p})\mathbf{k}. \tag{18}$$

This formally demonstrates that RoPE imparts translation invariance to the attention mechanism. $\square$

### A.4. A Fourier Perspective

The principle of constructing equivariant functions from irreducible representations is deeply connected to Fourier analysis. The Fourier transform provides a way to decompose any function on a group into a weighted sum (or integral) over its irreps. For the translation group on $\mathbb{R}^d$, these irreps are precisely the complex exponentials we used as our building blocks. Therefore, RoPE can be understood as a practical application of Fourier theory, using a discrete basis of Fourier modes (the chosen frequencies $\Omega$) to represent the positional signal.

**Definition A.6** (Fourier Transform on $\mathbb{R}^d$). The forward Fourier transform $\mathcal{F} : L^2(\mathbb{R}^d) \to L^2(\mathbb{R}^d)$ maps a function $f$ to its frequency-space representation $\hat{f}$. The coefficient for a frequency $\boldsymbol{\omega}$ is the projection of $f$ onto the corresponding irrep $\rho_{\boldsymbol{\omega}}$:

$$\hat{f}(\boldsymbol{\omega}) = \mathcal{F}\{f\}(\boldsymbol{\omega}) = \int_{\mathbb{R}^d} f(\mathbf{p})\overline{\rho_{\boldsymbol{\omega}}(\mathbf{p})}\, d\mathbf{p} = \int_{\mathbb{R}^d} f(\mathbf{p})e^{-i\boldsymbol{\omega}^\top \mathbf{p}}\, d\mathbf{p}. \tag{19}$$

The inverse transform reconstructs the function by integrating over all irreps:

$$f(\mathbf{p}) = \mathcal{F}^{-1}\{\hat{f}\}(\mathbf{p}) = \frac{1}{(2\pi)^d}\int_{\mathbb{R}^d} \hat{f}(\boldsymbol{\omega})\rho_{\boldsymbol{\omega}}(\mathbf{p})\, d\boldsymbol{\omega} = \frac{1}{(2\pi)^d}\int_{\mathbb{R}^d} \hat{f}(\boldsymbol{\omega})e^{i\boldsymbol{\omega}^\top \mathbf{p}}\, d\boldsymbol{\omega}. \tag{20}$$

## B. Equivariance Properties of Platonic Transformers

We formally establish the equivariance of our proposed architecture. We consider a point cloud $\{\mathbf{p}_i, \mathbf{v}_i, s_i\}_{i=1}^N$ consisting of positions, vectors, and scalars. A global rotation $R \in \mathcal{G}$ acts on these inputs as $\mathbf{p}_i \mapsto R\mathbf{p}_i$, $\mathbf{v}_i \mapsto R\mathbf{v}_i$, and $s_i \mapsto s_i$.

**Equivariant Feature Lifting.** Input features are first lifted to functions on the group $\mathcal{G}$. The lifting operator, Lift, maps the input point cloud to a set of feature maps $\{\mathbf{f}_i : \mathcal{G} \to \mathbb{R}^C\}_{i=1}^N$. Scalar components are copied to each frame, while vector components (from $\mathbf{p}_i, \mathbf{v}_i$) are lifted by projecting them onto each reference frame. This projection means expressing the vector's coordinates in the local basis of a given frame $\tilde{R} \in \mathcal{G}$, which is achieved by the transformation $\tilde{R}^{-1}\mathbf{v}$. This lifting procedure is equivariant by construction: a global rotation $R$ of the input point cloud results in the lifted feature maps transforming via the left regular representation, $L_R$. That is:

$$(\text{Lift}(R \cdot \text{cloud}))_i(\tilde{R}) = (\text{Lift}(\text{cloud}))_i(R^{-1}\tilde{R}) \triangleq (L_R \mathbf{f}_i)(\tilde{R}). \tag{21}$$

**Equivariant Linear Layers.** All linear layers $\Phi$ in our network are implemented as point-wise group convolutions, as shown in Eq. 8. These layers are equivariant to the action of the group by construction (Cohen et al., 2019, Thm. 3.1), satisfying $\Phi(L_R\mathbf{f}_i) = L_R(\Phi(\mathbf{f}_i))$.

This leads to our main proposition regarding the attention mechanism.

**Proposition B.1** (Equivariant Attention). *Let the queries* $Q_i$, *keys* $K_i$, *and values* $V_i$ *be equivariant feature maps produced by the equivariant linear layers. The RoPE-enhanced attention mechanism (Eq. 7), which computes outputs* $\mathbf{y}_i$, *is an equivariant operation. That is, if the inputs transform as* $\mathbf{f}_i \mapsto L_R\mathbf{f}_i$, *the outputs transform as* $\mathbf{y}_i \mapsto L_R\mathbf{y}_i$.

*Proof.* Let the inputs to the attention layer $(Q_i, K_i, V_i)$ transform under a global rotation $R$ as $Q_i' = L_R Q_i$, $K_i' = L_R K_i$, and $V_i' = L_R V_i$. We analyze the transformation of each component of the attention calculation.

The score function $s_{ij}(\tilde{R})$, which depends on $Q_i(\tilde{R})$ and $K_j(\tilde{R})$ (and potentially RoPE terms derived from lifted positions), will transform as:

$$s_{ij}'(\tilde{R}) = \text{score}(Q_i'(\tilde{R}), K_i'(\tilde{R}), \dots) = \text{score}(Q_i(R^{-1}\tilde{R}), K_j(R^{-1}\tilde{R}), \dots) = s_{ij}(R^{-1}\tilde{R}).$$

This means the score function itself is equivariant, $s_{ij}' = L_R s_{ij}$. Since the softmax operator is applied point-wise for each frame $\tilde{R}$ over the index $j$, the attention weights also transform equivariantly:

$$\text{attn}_{ij}'(\tilde{R}) = \text{softmax}_j(s_{ij}'(\tilde{R})) = \text{softmax}_j(s_{ij}(R^{-1}\tilde{R})) = \text{attn}_{ij}(R^{-1}\tilde{R}).$$

Finally, the output feature map $\mathbf{y}_i$ transforms as:

$$\mathbf{y}_i'(\tilde{R}) = \sum_{j=1}^{N} \text{attn}_{ij}'(\tilde{R}) V_j'(\tilde{R})$$

$$= \sum_{j=1}^{N} \text{attn}_{ij}(R^{-1}\tilde{R}) V_j(R^{-1}\tilde{R}) = \mathbf{y}_i(R^{-1}\tilde{R}).$$

Thus, the output transforms as $\mathbf{y}_i' = L_R \mathbf{y}_i$, proving the attention mechanism is equivariant. $\qquad\square$

## C. Proofs

### C.1. Proof of Proposition 4.1

We seek to show that the unnormalized attention score, which defines the kernel $\phi_{\mathbf{q}_i}(\Delta\mathbf{p})$, takes the form of a sparse Fourier series whose coefficients are linear projections of the query $\mathbf{q}_i$.

Let the relative position be $\Delta\mathbf{p} = \mathbf{p}_j - \mathbf{p}_i$. With a constant key vector $\mathbf{k}_j = \mathbf{1}$, the kernel is defined by the attention score:

$$\phi_{\mathbf{q}_i}(\Delta\mathbf{p}) = (\boldsymbol{\rho}(\mathbf{p}_i)\mathbf{q}_i)^\top (\boldsymbol{\rho}(\mathbf{p}_j)\mathbf{1})$$

Using the properties of the RoPE operator $\boldsymbol{\rho}$, this simplifies to:

$$\phi_{\mathbf{q}_i}(\Delta\mathbf{p}) = \mathbf{q}_i^\top \boldsymbol{\rho}(\mathbf{p}_i)^\top \boldsymbol{\rho}(\mathbf{p}_j)\mathbf{1} = \mathbf{q}_i^\top \boldsymbol{\rho}(\Delta\mathbf{p})\mathbf{1}$$

The RoPE matrix $\boldsymbol{\rho}(\Delta\mathbf{p})$ is block-diagonal, consisting of $d/2$ independent 2D rotation blocks. We can therefore analyze the contribution from a single block $k$ and sum the results. Let $\theta_k = \boldsymbol{\omega}_k^\top \Delta\mathbf{p}$. The contribution from block $k$ is:

$$\phi_k = \begin{pmatrix} q_{2k-1} & q_{2k} \end{pmatrix} \begin{pmatrix} \cos(\theta_k) & -\sin(\theta_k) \\ \sin(\theta_k) & \cos(\theta_k) \end{pmatrix} \begin{pmatrix} 1 \\ 1 \end{pmatrix}$$

Performing the matrix-vector multiplications, we get:

$$\phi_k = \begin{pmatrix} q_{2k-1} & q_{2k} \end{pmatrix} \begin{pmatrix} \cos(\theta_k) - \sin(\theta_k) \\ \sin(\theta_k) + \cos(\theta_k) \end{pmatrix}$$

$$= q_{2k-1}(\cos(\theta_k) - \sin(\theta_k)) + q_{2k}(\sin(\theta_k) + \cos(\theta_k))$$

Grouping terms by $\cos(\theta_k)$ and $\sin(\theta_k)$ reveals the linear projections:

$$\phi_k = \underbrace{(q_{2k-1} + q_{2k})}_{a_k(\mathbf{q}_i)} \cos(\theta_k) + \underbrace{(q_{2k} - q_{2k-1})}_{b_k(\mathbf{q}_i)} \sin(\theta_k)$$

The Fourier coefficients $a_k(\mathbf{q}_i)$ and $b_k(\mathbf{q}_i)$ are thus simple linear combinations of the query vector's elements. Summing over all $k = 1, \dots, d/2$ yields the complete kernel $\phi_{\mathbf{q}_i}(\Delta\mathbf{p})$, which has the exact form stated in the proposition.

### C.2. Proof of Corollary 4.3

The linear-time complexity is achieved by expressing the operation in matrix form and re-ordering the computation. Let $\mathbf{q}'_i = \boldsymbol{\rho}_\Omega(\mathbf{p}_i)\mathbf{q}_i$ and $\mathbf{k}'_j = \boldsymbol{\rho}_\Omega(\mathbf{p}_j)\mathbf{1}$. Let $\mathbf{Q}' \in \mathbb{R}^{N \times d'}$ be the matrix with rows $(\mathbf{q}'_i)^\top$, $\mathbf{K}' \in \mathbb{R}^{N \times d'}$ be the matrix with rows $(\mathbf{k}'_j)^\top$, and $\mathbf{V} \in \mathbb{R}^{N \times d_v}$ be the matrix of value vectors. The output matrix $\mathbf{Y} \in \mathbb{R}^{N \times d_v}$ is given by:

$$\mathbf{Y} = (\mathbf{Q}'(\mathbf{K}')^\top)\mathbf{V}.$$

By the associativity of matrix multiplication, this can be computed as $\mathbf{Y} = \mathbf{Q}'((\mathbf{K}')^\top \mathbf{V})$. The term $(\mathbf{K}')^\top \mathbf{V}$ costs $O(Nd'd_v)$ to compute, resulting in a $d' \times d_v$ matrix. Multiplying this by $\mathbf{Q}'$ costs an additional $O(Nd'd_v)$. The total complexity is therefore $O(Nd'd_v)$, linear in the sequence length $N$.

### C.3. Proof of Platonic Transformers Implementing Group Convolutions

The dynamic convolution from Proposition 4.1 becomes a dynamic *group convolution* within the Platonic Transformer. This is a direct consequence of applying the operation to lifted coordinates $\mathbf{p}_i(R) = R^{-1}\mathbf{p}_i$ for each reference frame $R \in \mathcal{G}$. Since the relative position vector becomes $R^{-1}(\mathbf{p}_j - \mathbf{p}_i)$, the kernel's input is transformed accordingly. The resulting output for each frame takes the form of a group cross-correlation[3]:

$$\mathbf{y}_i(R) = \sum_{j=1}^{N} \phi_{\mathbf{q}_i(R)}\left(R^{-1}(\mathbf{p}_j - \mathbf{p}_i)\right)\mathbf{v}_j(R). \tag{22}$$

Here, the kernel $\phi_{\mathbf{q}_i(R)}$ is steered by the group element $R$, defining an equivariant dynamic group convolution.

## D. Equivalent Attention via RoPE Base Frequency Steering

We achieve full equivariance to Euclidean transformations by making the RoPE operator dependent on a local reference frame $R$ by projecting positions $\mathbf{p}_i$ on $R$ to obtain positions $\mathbf{p}_i(R) := R^{-1}\mathbf{p}_i$. The attention scores $s_{ij}(R)$ for a query $\mathbf{q}(R)_i$ and key $\mathbf{k}(R)_j$ are computed as:

$$s_{ij}(R) = \mathbf{q}_i(R)^\top \boldsymbol{\rho}_\Omega((\mathbf{p}_j - \mathbf{p}_i)(R))\mathbf{k}_j(R), \tag{23}$$

An equivalent approach is to steer the set of base RoPE frequencies $\Omega$ for each frame, creating a frame-specific set $\Omega_R = \{R\boldsymbol{\omega}_k \mid \boldsymbol{\omega}_k \in \Omega\}$ (Reddy & Chatterji, 1996). The attention scores are then computed as:

$$\hat{s}_{ij}(R) = \mathbf{q}_i(R)^\top \boldsymbol{\rho}_{\Omega_R}(\mathbf{p}_j - \mathbf{p}_i)\mathbf{k}_j(R). \tag{24}$$

*Proof.* For $s_{ij}(R)$ and $\hat{s}_{ij}(R)$ to be equivalent, we require that $\rho_\Omega((\mathbf{p}_j - \mathbf{p}_i)(R)) = \rho_{\Omega_R}(\mathbf{p}_j - \mathbf{p}_i)$. For this, we need to show that $\boldsymbol{\omega}_k^\top \Delta\mathbf{p}(R) = (R\boldsymbol{\omega})_k^\top \Delta\mathbf{p}$, where $\Delta\mathbf{p} = \mathbf{p}_j - \mathbf{p}_i$. Let $\mathbf{R}$ be the orthogonal matrix corresponding to $R$. Then we have:

$$\boldsymbol{\omega}_k^\top \Delta\mathbf{p}(R) = \boldsymbol{\omega}_k^\top R^{-1}\Delta\mathbf{p} \tag{25}$$
$$= \boldsymbol{\omega}_k^\top \mathbf{R}^\top \Delta\mathbf{p} \tag{26}$$
$$= (\mathbf{R}\boldsymbol{\omega}_k)^\top \Delta\mathbf{p} \tag{27}$$
$$= (R\boldsymbol{\omega}_k)^\top \Delta\mathbf{p} \tag{28}$$

Thus, projecting global positions or steering the base frequencies are equivalent. $\square$

By projecting the global positions, the RoPE attention mechanism remains identical to its traditional formulation. Steering the base frequencies, however, is often more computationally efficient, since the number of base frequencies is typically much smaller than the input sequence length.

---

[3]Following common convention, we refer to this operation as a group convolution, though it is technically a cross-correlation (Cohen & Welling, 2016; Bekkers, 2020).

# E. Details of Architecture

In this section, we provide additional details about the architecture of the Platonic Transformer and the various model configurations used in our experiments. Our framework is designed to be equivariant to roto-translation groups, primarily $SE(n)$ and, through specific configurations, the full Euclidean group $E(n)$.

We denote the core embedding dimension per group element as $d_{\text{hidden}}$. Since our features are functions on a group $\mathcal{G}$ of order $|\mathcal{G}|$, the total feature dimension of a layer is $d_{\text{model}} = |\mathcal{G}| \times d_{\text{hidden}}$. The specific group is determined by the `solid_name` parameter.

For the initial feature processing, input scalars and vectors are first embedded and then lifted into a group-equivariant feature space using an initial lifting operation. This creates a tensor where the channel dimension is expanded by a factor of $|\mathcal{G}|$. An initial group-equivariant linear layer then projects these lifted features to the model's working dimension, $d_{\text{model}}$. Optionally, an equivariant Absolute Positional Encoding (APE), parameterized by `ape_sigma`, can be added at this stage.

The main body of the network consists of a stack of equivariant transformer blocks. Each block contains two main sub-modules: a group-equivariant interaction layer and a feed-forward network (FFN), connected with residual connections. Normalization is applied either before each sub-module or after.

For the group-equivariant interaction layer, we denote the number of attention heads per group element as $n_{\text{head}}$. The total number of effective parallel heads is therefore $|\mathcal{G}| \times n_{\text{head}}$. The dimension of each head, $d_{\text{head}}$, is calculated as $d_{\text{hidden}}/n_{\text{head}}$. The input features are first projected to query, key, and value representations using group-equivariant linear layers. To encode relative spatial information, group-equivariant Rotary Position Embeddings, parameterized by `rope_sigma` and `learned_freqs`, are applied to the query, key and value vectors. The interaction can then be performed either as a full softmax-based attention mechanism or as a linear-time dynamic group convolution via the `attention` flag. For the Feed Forward Networks (FFNs), we denote the hidden feature dimension as $d_{\text{ffn}} = d_{\text{model}} \times f_{\text{factor}}$. The FFN consists of two group-equivariant linear layers with a GELU activation function in between.

For the final output, two separate readout heads project the features to the desired scalar and vector output dimensions. For graph-level tasks, a pooling operation performs a mean aggregation over the node and group dimensions to produce a final invariant prediction. For node-level tasks, an averaging operation over the group axis projects the features back to standard invariant scalar and equivariant vector representations. Following standard Transformer practices, we apply dropout to the attention weights and FFN activations, and stochastic depth to the outputs of the equivariant transformer blocks.

Particular values for all the important hyperparameters used for the experiments are in the Table.12

# F. Hyperparameter Tuning and Model Selection Strategy

This section outlines the full procedure used to configure and train our models.

**Baseline Optimization** To establish a fair point of comparison, we first optimized the general training protocol using only the translation-only equivariant ($T(n)$) models. This initial phase involved tuning the optimizer, learning rate schedule, weight decay, and data augmentations to ensure the baseline models were as competitive as possible. This fixed protocol was then used for all subsequent experiments.

**Hyperparameter Sweep for Model Selection** With the training protocol fixed, we performed an extensive hyperparameter sweep for both $SE(n)$ and $T(n)$ model classes. This sweep was designed to find the optimal architectural parameters while maintaining an equal computational budget between model families. The parameters and their swept values are summarized in Table 7.

The hyperparameter sweep was conducted across multiple layers and three random seeds for a *moderate number of epochs* to efficiently explore the configuration space. It should be noted that some configurations are not applicable for the Octahedral group, as its 24 symmetry elements require a minimum of 24 total effective heads (i.e., at least one head per group element). After identifying the best-performing hyperparameters for both the $SE(n)$ and $T(n)$ model families from this sweep, we proceeded to a final, full-length training run. These selected models were trained for a *large number of epochs* to ensure convergence again with fixed compute budget, producing the final results reported in the main paper.

*Table 7.* Hyperparameter Sweep Configurations.

| Parameter | Configuration | Values |
|---|---|---|
| Hidden Dim | - | $[384, 576, 768, 1152, 1920]$ |
| Layers | - | $[7 - 20]$ |
| Number of Heads | $T(n)$ **model** (HS=16) $T(n)$ **model** (HS=32) $SE(n)$ **model** (HS=16) $SE(n)$ **model** (HS=32) | $[24, 36, 48, 72]$ $[12, 18, 24, 36]$ $[2, 3, 4, 6]$ heads per group element $[1, 2, 3]$ heads per group element |
| Rope Sigma ($\sigma_{\text{rope}}$) | RoPE frequency scaling | $[0.5 - 2.0]$ |
| Attention | Full Attention / Linear Conv | $[\text{True}, \text{False}]$ |
| Solid Group ($\mathcal{G}$) | Symmetry group | $[\text{Octahedron}, \text{Tetrahedron}, C_{2-8}, D_{4-8}]$ |
| Lambda F ($\lambda_F$) | OMol Force loss weight | $[1.0 - 25.0]$ |
| Lambda E ($\lambda_F$) | OMol Energy loss weight | $[1.0 - 15.0]$ |
| Batch Size | Samples/Atoms per batch | $[64 - 512]/[3000 - 24000]$ |
| Weight Decay | | $[1e^{-3} - 1e^{-7}]$ |

## F.1. Heads vs. Group Size

We present a targeted ablation over (i) the effective number of heads, (ii) the effective head dimensionality, and (iii) the group size would clarify whether performance gains come primarily from adding more geometric frames or from increasing feature diversity per frame. In our current experiments, head size is fixed and we implicitly ablate group size across datasets. Overall, increasing the number of frames improves performance, but with diminishing returns. One plausible explanation is that as group size grows, the effective head dimensionality per frame becomes too small, limiting capacity. Finally, we emphasize that for downstream use we would not necessarily maximize the number of group elements under a fixed compute budget; instead, we would select group size and head configuration via task-specific ablations and resource constraints.

*Table 8.* QM9 ablation studying the trade-off between group size (number of geometric frames) and feature diversity per frame. We report hidden dimension $d_{\text{model}}$, per-frame dimension $d_{\text{model}}/|\mathcal{G}|$, the configured number of heads, and the resulting effective number of heads and per-head dimension.

| Group | $d_{\text{model}}$ | $d_{\text{model}}/|\mathcal{G}|$ | #Heads | Eff. #Heads | Eff. head dim | $\alpha$ | $\mu$ |
|---|---|---|---|---|---|---|---|
| 1 (Trivial) | 1152 | 1152 | 72 | 72 | 16 | 0.028 | 0.064 |
| 12 (Tetrahedron) | 1152 | 96 | 72 | 6 | 16 | 0.012 | 0.049 |
| 24 (Octahedron) | 1152 | 48 | 72 | 3 | 16 | 0.010 | 0.048 |
| 1 (Trivial) | 1152 | 1152 | 48 | 48 | 24 | 0.027 | 0.064 |
| 12 (Tetrahedron) | 1152 | 96 | 48 | 4 | 24 | 0.012 | 0.048 |
| 24 (Octahedron) | 1152 | 48 | 48 | 2 | 24 | 0.011 | 0.047 |

# G. Additional Results on ImageNet-1K

To further demonstrate the efficacy of the equivariance constraint, we provide additional results on ImageNet-1K (Deng et al., 2009). As shown in Table 9, we observe the proposed constraint consistently outperforms the trivial unconstrained baseline or matches performance while requiring fewer parameters and shorter compute time. Despite ImageNet-1K falling under non-equivariant tasks, the constraint comes with computational advantages. ImageNet-1K is a large-scale, non-aligned dataset, and while our models are not optimized for state-of-the-art performance, our results demonstrate that our Flop-2d$_2$ model achieves parity with the trivial ViT baseline while halving the parameter count. This efficiency gain enables two distinct deployment strategies: prioritizing efficient training, or scaling model capacity (e.g., the wide configuration) to match the baseline's computational footprint, which provides clear performance improvement.

*Table 9.* ImageNet-1K results demonstrating the utility of equivariance constraints in the Platonic Transformer used in a ViT setup. We outperform the Trivial variant while maintaining a modest parameter budget.

| Model | Hidden dim. | # Params | Best Top-1 | Epoch Time (min) |
|---|---|---|---|---|
| Trivial | 768 | 78.9M | 79.53% | 9.4 |
| Flop-2d | 768 | 39.9M | 79.17% | 7.6 |
| Flop-2d (wide) | 1088 | 79.5M | 80.41% | 9.4 |

# H. Regression Experiments on QM9

## H.1. Description of the dataset

The QM9 dataset (Ramakrishnan et al., 2014) contains up to 9 heavy atoms and 29 atoms, including hydrogens. We use the train/val/test partitions introduced in Gilmer et al. (2017), which consist of 100K/18K/13K samples, respectively, for each partition.

## H.2. Training Details for the Regression Experiment

For the QM9 regression task, we train the Platonic Transformer to predict molecular properties. Before being fed to the model, the input molecular geometries are centered by subtracting the mean coordinate of each molecule. To stabilize training, we normalize the target property values by subtracting their mean and dividing by their standard deviation, with these statistics computed over the training set. We employ data augmentation in the form of random $SO(3)$ rotations applied to the coordinates during training.

The model is trained for a total of 1000 epochs using a batch size of 96. We utilize the Adam optimizer with a learning rate of $5 \times 10^{-4}$ and a weight decay of $10^{-8}$. A cosine annealing schedule with a 10-epoch linear warmup adjusts the learning rate throughout training. To prevent exploding gradients, we apply gradient clipping with a maximum norm of 0.5. The training objective is the Mean Absolute Error (MAE) on the normalized target values, while validation and testing are performed by calculating the MAE on the original, unnormalized scale.

At test time, we optionally apply test-time augmentation (TTA) by averaging the original prediction with four additional predictions obtained from independently sampled random $SO(3)$ rotations of each molecule, giving five test evaluations in total. The single-orientation prediction is also logged separately as the w/o TTA score. Since molecular targets are invariant to global rotations, this averaging should not change the ideal prediction, but it reduces residual orientation-dependent numerical noise in finite-precision equivariant computation. Further hyperparameter details are available in Table 12.

## H.3. Regression results on QM9

Table 10 summarizes the results on regression on QM9. The comparison shows a consistent benefit from the octahedral equivariance constraint over the translation-only Trivial variant: the Octahedron-Attention model improves on the corresponding Trivial-Attention model for every reported target, with especially clear gains on electronic properties such as $\Delta\varepsilon$, $\varepsilon_{\mathrm{HOMO}}$, $\varepsilon_{\mathrm{LUMO}}$, and $\mu$. In absolute terms, the model is competitive with highly specialized equivariant architectures, obtaining the strongest reported errors among the listed methods for $\Delta\varepsilon$ and $\mu$, while remaining close to the best entries for several other targets. The results are nevertheless not uniformly dominant: established molecular architectures such as PaiNN (Schütt et al., 2021), TorchMD-NET (Thölke & Fabritiis, 2022), and SphereNet (Liu et al., 2022) remain stronger on some thermochemical and spatial targets. Thus, the table is best read as evidence that the Platonic Transformer provides a strong general-purpose geometric prior, rather than as a target-wise replacement for heavily tuned molecular models.

The table also reports both test-time augmentation (TTA) and single-orientation evaluation. Empirically, the gains are modest on several electronic targets and larger on some extensive thermochemical quantities and $R^2$; for example, Octahedron-Attention improves from .047 to .043 on $\alpha$, from .0097 to .0087 on $\mu$, from 11.3 to 8.53 on $G$, and from .212 to .138 on $R^2$.

Crucially, unlike baselines such as EquiformerV2 which rely on target-specific hyperparameter tuning, we employ a single fixed set of hyperparameters across all targets. Despite this constraint, the Platonic Transformer achieves competitive results, suggesting that further performance gains could be realized with target-specific optimization.

*Table 10.* Mean absolute error results on QM9 test set. † denotes using different data partitions.

| Model | Task Units | $\alpha$ $a_0^3$ | $\Delta\varepsilon$ meV | $\varepsilon_{\text{HOMO}}$ meV | $\varepsilon_{\text{LUMO}}$ meV | $\mu$ D | $C_\nu$ cal/mol K | $G$ meV | $H$ meV | $R^2$ $a_0^2$ | $U$ meV | $U_0$ meV | ZPVE meV |
|---|---|---|---|---|---|---|---|---|---|---|---|---|---|
| DimeNet++ (Gasteiger et al., 2020) | | .044 | 33 | 25 | 20 | .030 | .023 | 8 | 7 | .331 | 6 | 6 | 1.21 |
| EGNN (Satorras et al., 2021)† | | .071 | 48 | 29 | 25 | .029 | .031 | 12 | 12 | .106 | 12 | 11 | 1.55 |
| PaiNN (Schütt et al., 2021) | | .045 | 46 | 28 | 20 | .012 | .024 | **7.35** | **5.98** | .066 | **5.83** | **5.85** | 1.28 |
| TorchMD-NET (Thölke & Fabritiis, 2022) | | .059 | 36 | 20 | 18 | .011 | .026 | 7.62 | 6.16 | **.033** | 6.38 | 6.15 | 1.84 |
| SphereNet (Liu et al., 2022) | | .046 | 32 | 23 | 18 | .026 | **.021** | 8 | 6 | .292 | 7 | 6 | **1.12** |
| SEGNN (Brandstetter et al., 2022)† | | .060 | 42 | 24 | 21 | .023 | .031 | 15 | 16 | .660 | 13 | 15 | 1.62 |
| EQGAT (Le et al., 2022) | | .053 | 32 | 20 | 16 | .011 | .024 | 23 | 24 | .382 | 25 | 25 | 2.00 |
| Equiformer (Liao & Smidt, 2023) | | .046 | 30 | 15 | 14 | .011 | .023 | 7.63 | 6.63 | .251 | 6.74 | 6.59 | 1.26 |
| EquiformerV2 (Liao et al., 2024) | | .050 | 29 | **14** | **13** | .010 | .023 | 7.57 | 6.22 | .186 | 6.49 | 6.17 | 1.47 |
| PΘNITA (Bekkers et al., 2024) | | **.038** | 30.4 | 16.0 | 14.5 | .012 | .024 | 8.63 | 8.04 | .235 | 8.67 | 8.31 | 1.29 |
| Platonic Transformer (Trivial, Attn, TTA) | | .059 | 38.6 | 23.3 | 20.2 | .017 | .030 | 13.2 | 13.4 | .191 | 13.8 | 13.4 | 1.57 |
| Platonic Transformer (Trivial, Conv, TTA) | | .062 | 44.9 | 26.5 | 23.9 | .024 | .033 | 13.3 | 12.9 | .150 | 13.2 | 12.7 | 1.61 |
| Platonic Transformer (Octa, Attn, TTA) | | .043 | **28.98** | 15.8 | 13.4 | **.0087** | .021 | 8.53 | 8.26 | .138 | 8.77 | 9.29 | 1.36 |
| Platonic Transformer (Octa, Conv, TTA) | | .045 | 31.8 | 16.7 | 15.1 | .011 | .024 | 14.1 | 18.8 | .138 | 11.2 | 22.0 | 1.40 |
| Platonic Transformer (Trivial, Attn, w/o TTA) | | .060 | 38.8 | 23.4 | 20.3 | .018 | .030 | 14.7 | 15.1 | .275 | 15.6 | 15.4 | 1.66 |
| Platonic Transformer (Trivial, Conv, w/o TTA) | | .065 | 45.5 | 26.9 | 24.9 | .027 | .035 | 14.7 | 14.4 | .205 | 14.9 | 14.1 | 1.69 |
| Platonic Transformer (Octa, Attn, w/o TTA) | | .047 | 30.0 | 16.5 | 14.0 | .0097 | .023 | 11.3 | 11.4 | .212 | 12.3 | 13.1 | 1.58 |
| Platonic Transformer (Octa, Conv, w/o TTA) | | .050 | 33.2 | 17.3 | 16.1 | .012 | .026 | 16.3 | 23.6 | .191 | 13.6 | 28.3 | 1.40 |

*Table 11.* Runtime comparison. Left: inference wall-clock times, Right: mean training-step timing on input, decomposed into forward, backward, and optimizer-step time on QM9.

*(a)* QM9 inference wall-clock times.

| Platonic Transformer | |
|---|---|
| **Group** | Avg. Time (ms) (↓) |
| {e} | $2.87 \pm 0.29$ |
| Tetrahedron | **2.79** $\pm$ 0.21 |
| Octahedron | $2.85 \pm 0.25$ |
| Reference methods | |
| **Method** | Avg. Time (ms) (↓) |
| Standard Transformer | **2.01** $\pm$ 3.74 |
| G-Hyena [46] | $44.06 \pm 60.05$ |
| TFN [64] | $590.45 \pm 269.25$ |

*(b)* 3D point-cloud (QM9) training-step timing.

| Solid | Params | Forward | Backward | Opt. step | Total |
|---|---|---|---|---|---|
| Trivial | 224.99M | 33.44 (0.96×) | 31.55 (0.93×) | 1.92 (2.23×) | 66.91 (0.96×) |
| Tetrahedron | 18.86M | 34.18 (0.98×) | 30.25 (0.89×) | 0.98 (1.14×) | 65.41 (0.94×) |
| Octahedron | 9.49M | 34.98 (1.00×) | 33.83 (1.00×) | 0.86 (1.00×) | 69.68 (1.00×) |
| Icosahedron | 3.87M | 34.43 (0.98×) | 41.71 (1.23×) | 0.61 (0.71×) | 76.75 (1.10×) |

## H.4. Runtime measurements

Given that the Platonic Transformer preserves the standard Transformer computation graph, our method achieves inference speeds of the same order of magnitude as a standard Transformer layer. As shown in Table 11a, on QM9 a single Platonic Transformer layer runs in roughly 2.8 ms on a batch of 64 molecules on a single H200 GPU, averaged over 10 batches. This is substantially faster than the geometric reference methods considered here, with G-Hyena and Tensor Field Networks requiring 44.06 ms and 590.45 ms per layer, respectively, under the same setup. To produce the QM9 wall-clock timing for the standard Transformer in Table 11a, node features from $B = 64$ QM9 molecules were projected from $d_{\text{in}} = 11$ to $d_{\text{model}} = 512$ using a linear layer and fed as tokens into a `TransformerEncoderLayer` module provided by PyTorch with 16 heads. We measure wall-clock timings for a forward pass over 10 batches on a single H200 GPU. This provides a reference timing for comparing the inference speed of the Platonic Transformer with other geometric baselines such as G-Hyena (Moskalev et al., 2025) and Tensor Field Networks (Thomas et al., 2018).

Table 11b also reports a training-step wall-clock breakdown of different symmetry group on a representative QM9 batch input. We benchmark on a single NVIDIA H100 GPU (100 GB, CUDA 12.6) using FlashAttention, `torch.compile`, and fused SGD. The batch contains $B = 167$ molecules (mean $\approx 18$ atoms/molecule; $\approx 3006$ atoms per batch) with $d_{\text{model}} = 1200$; we report mean times over 200 measured steps after 50 warmup steps. The forward pass is essentially unchanged across variants (33.44–34.98 ms) and the backward pass is comparable for the tetrahedral and octahedral models (30.25 ms and 33.83 ms vs. 31.55 ms for the trivial model), while the icosahedral model incurs a higher backward cost (41.71 ms). In contrast, equivariant weight sharing substantially reduces the parameter count (224.99M → 3.87M), making the optimizer step cheaper (1.92 ms for the trivial model vs. 0.98/0.86/0.61 ms for tetrahedral/octahedral/icosahedral). Overall step times

(excluding `zero_grad`) are 66.91 ms (trivial), 65.41 ms (tetrahedral), 69.68 ms (octahedral), and 76.75 ms (icosahedral), supporting the claim that equivariance can be introduced at comparable wall-clock cost while reducing parameter-update overhead.

# I. Details of experiments on Cifar10

## I.1. Description of the dataset

The CIFAR-10 dataset (Krizhevsky, 2009) is a standard benchmark for image classification, consisting of 60,000 32x32 color images across 10 classes. The dataset is divided into a training set of 50,000 images and a test set of 10,000 images.

## I.2. Training Details

For the CIFAR-10 classification task, our experimental setup is closely adapted from the supervised training recipe for Vision Transformers presented in DeiT-III (Touvron et al., 2022). We tokenize each image into a sequence of non-overlapping patches using a patch size of $4 \times 4$ pixels, a key deviation from the ImageNet configurations to suit the lower resolution of the dataset.

The model is trained using the LAMB optimizer, which is subject to a cosine decay schedule following a 5-epoch warm-up period. A comprehensive suite of regularization techniques is employed, including a weight decay of 0.02, Mixup with an alpha value of 0.8, and CutMix with an alpha of 1.0, in addition to model-size-dependent Stochastic Depth. The data augmentation pipeline is built upon the '3-Augment' strategy, incorporating standard Random Resized Crop (RRC), horizontal flips, ColorJitter with a factor of 0.3, and a single, randomly selected transformation from a pool of three: Grayscale, Solarization, or Gaussian Blur.

The training objective is optimized using a Binary Cross-Entropy (BCE) loss, and positional information is supplied to the transformer blocks through a combination of both Absolute Positional Encodings (APE) and Rotary Position Embeddings (RoPE). Further hyperparameter details are available in Table 12.

# J. Details of experiments on ScanObjectNN

## J.1. Description of the dataset

ScanObjectNN(Uy et al., 2019) dataset is a real-world 3D point cloud dataset. It contains 15,000 objects divided into 15 categories with 2902 unique object instances. It contains background, parts missing, and object deformation elements, which makes the classification task a challenge. The dataset consists of three variants OBJ_BG, OBJ_ONLY and PB_T50_RS, for now the latter is only examined.

## J.2. Training Details

In order to prepare the input point cloud $\mathbf{P} \in \mathbb{R}^{N \times 3}$ for processing by the Platonic Transformer, we follow a preprocessing procedure. Similar to established methods (Pang et al., 2023; Yu et al., 2022), we first use Farthest Point Sampling (FPS) to select a set of $L = 2048$ central points, denoted as $\mathbf{P}_C \in \mathbb{R}^{L \times 3}$ with $L = 2048$. Subsequently, for each central point $P_C^i$, we define a local patch $x_p^i \in \mathbb{R}^{K \times 3}$ by identifying its K-Nearest Neighbors (KNN) within the original point cloud P. These local patches serve as the primary input vectors to the Platonic Transformer.

Additionally, to account for the axis-aligned nature of the dataset and to provide the model with a global reference frame, we incorporate rotation augmentation. For each input vector, a rotation matrix is applied. This matrix is either a random rotation or the 3×3 identity matrix, which is concatenated with the input vector to provide the model with information about the global orientation.

# K. Details of experiments on OMol25

## K.1. Description of the dataset

For large-scale molecular experiments, we use the Open Molecules 2025 (OMol25) dataset (Levine et al., 2025), a comprehensive collection of over 100 million Density Functional Theory (DFT) calculations performed at the wB97M-

*Table 12.* Hyperparameters for all datasets

| Hyperparameter | QM9 | OMol25 | CIFAR10 | ScanObjectNN | ProteinMD |
|---|---|---|---|---|---|
| **Architecture** | | | | | |
| hidden_dim | 1152 | 1920 | 768 | 768 | 1152 |
| num_layers | 14 | 16 | 12 | 12 | 5 |
| num_heads | 72 | 12 | 12 | 48 | 72 |
| **Positional encoding** | | | | | |
| rope_sigma | 1.5 | 2.0 | 16.0 | 18.0 | 1 |
| ape_sigma | 0.5 | None | 16.0 | 10.0 | None |
| learned_freqs | True | True | True | True | True |
| freq_init | spiral | random | spiral | spiral | random |
| **Attention / readout** | | | | | |
| attention | True | True | True | True | False |
| use_key | False | True | False | False | False |
| qk_norm | False | True | False | False | False |
| rope_on_values | True | True | False | True | False |
| dropout | 0.0 | 0.0 | 0.0 | 0.0 | 0 |
| drop_path_rate | 0.0 | 0.0 | 0.1 | 0.0 | 0 |
| mean_aggregation | False | False | False | False | False |
| ffn_dim_factor | 4 | 4 | 4 | 4 | – |
| layer_scale_init_value | None | 1e-4 | None | None | None |
| **Training** | | | | | |
| train_augm | True | True | False | True | False |
| lr | 5e-4 | 5e-4 | 8e-4 | 2e-4 | 5e-4 |
| batch_size | 96 | 3000[†] | 256 | 64 | 32 |
| epochs | 1000 | 20 | 500 | 300 | 20 |
| warmup | 10 | 1% steps | 20 | 10 | 5 |
| weight_decay | 1e-8 | 1e-8 | 0.05 | 1.5e-5 | 1e-8 |
| lambda_F | – | 20.0 | – | – | – |
| lambda_E | – | 10.0 | – | – | – |
| cosine_scheduler | True | True | True | True | True |
| gpus | 1 | 4 | 1 | 1 | 1 |

[†] OMol uses dynamic batching: each step packs up to 12000 atoms / 2.4M edges across 4GPUs.

V/def2-TZVPD level of theory. This dataset is notable for its vast chemical and structural diversity, encompassing 83 elements and systems up to 350 atoms. The structures are drawn from a wide range of chemical domains, including small molecules, biomolecules, metal complexes, and electrolytes, and feature varied charges, spin states, conformers, and reactive geometries.

The OMol25 dataset is organized into several training sets and splits for validation and testing to ensure consistent and robust model evaluation. The full training set, "All," contains over 100 million DFT calculations. For more computationally efficient training and development, a smaller, uniformly sampled "4M" split is provided, containing approximately 4 million structures. Our work primarily utilizes the "Neutral" split, which consists of approximately 34 million charge-neutral, singlet structures drawn from established community datasets like ANI-2X, GEOM, and SPICE2. This split is designed to benchmark model performance on familiar organic chemistry space without the added complexity of variable charge and spin.

For validation and testing, OMol25 provides several out-of-distribution (OOD) splits designed to evaluate model generalizability. The primary validation set ("Val Comp") consists of structures with compositions held out from the training set. Further specialized test sets include held-out organic and metal-complex reactions ("Test Reactivity"), experimental crystal structures from the Crystallography Open Database ("Test COD"), and unique anion structures ("Test Anions"), among others. The core task is Structure to Energy and Forces (S2EF), where models are evaluated on their ability to predict the total energy of a structure and the per-atom forces, with Mean Absolute Error (MAE) being the primary metric.

### K.2. Training Details

For OMol25, we train on the 4M training split and evaluate on the held-out validation split provided with the dataset. The model is trained using AdamW with a learning rate of $5 \times 10^{-4}$ and weight decay $10^{-8}$. We use a cosine decay schedule with a linear warmup over the first $1\%$ of optimizer steps and a minimum learning rate of $10^{-6}$. Training is conducted for 20 epochs. To stabilize optimization, we apply gradient clipping with a maximum norm of $1.0$ and maintain an exponential moving average of the model weights with decay $0.99$ after a warmup of 2000 steps. We also use random $O(3)$ data augmentation, applying random rotations and reflections to the molecular geometries and forces during training.

We train with dynamic batching, where each batch is packed up to a fixed computational budget rather than a fixed number of molecules. Specifically, each optimizer step contains at most 12,000 atoms and 2.4M edges. This matches the effective batch size used in our distributed training setup while allowing batches to adapt to the varying molecule sizes in OMol25. Validation is performed every 5000 optimizer steps and is capped at 500 validation batches.

The training objective is a weighted sum of two components: a per-atom energy MAE and an L2-norm MAE on the force vectors. The force loss is calculated as the average Euclidean norm of the error between predicted and target force vectors. The total loss is

$$\mathcal{L} = \lambda_E \mathcal{L}_E + \lambda_F \mathcal{L}_F, \tag{29}$$

with $\lambda_E = 10.0$ and $\lambda_F = 20.0$.

To ensure stable training on this large-scale task, we normalize the target energies using a linear referencing scheme. We subtract precomputed elemental reference energies from the raw DFT total energy:

$$E_{\text{ref}} = E_{\text{DFT}} - \sum_{i=1}^{N} E_{Z_i}^{\text{atom}}, \tag{30}$$

where $E_{\text{ref}}$ is the referenced target energy, $E_{\text{DFT}}$ is the system's total DFT energy, $N$ is the number of atoms, $Z_i$ is the atomic number of atom $i$, and $E_{Z_i}^{\text{atom}}$ is the precomputed reference energy for that element. Energies and forces are then scaled by the training-set RMSD. This procedure is consistent with the methodology used for the OC22 dataset (Tran et al., 2023) and helps maintain comparability with other large-scale models.

For the OMol25 experiments, we use learned keys together with QK normalization, RoPE on values, charge/spin conditioning, and FlashAttention. The detailed hyperparameters for this configuration are summarized in Table 12.

# L. Details of experiments on ProteinMD

## L.1. Description of the dataset

ProteinMD is a molecular dynamics benchmark derived from protein trajectories processed with MDAnalysis (Han et al., 2022). The task is to predict atomic force vectors from protein conformations, making it a large-scale geometric regression problem where both local bonded interactions and longer-range spatial interactions are important. Following prior work, we evaluate on two variants of the AdK equilibrium molecular dynamics trajectory (Seyler & Beckstein, 2017): a backbone-level system with 855 atoms and an all-atom system with 3,341 atoms. The dataset contains 4,186 protein structures with trajectories. We report force mean squared error (MSE), consistent with the evaluation protocol used by the reference methods in Table 6.

## L.2. Training Details

For ProteinMD, we use the linear convolutional variant of the Platonic Transformer, which is better suited to the long protein sequences in this benchmark than quadratic full attention. The model has 5 layers, hidden dimension 1152, and 72 heads. We use learned RoPE frequencies with `rope_sigma = 1`, no absolute positional encoding, and fixed keys. We do not apply additional rotation augmentation for this task.

The model is trained for 20 epochs with batch size 32 using AdamW with learning rate $5 \times 10^{-4}$ and weight decay $10^{-8}$. The learning rate follows a cosine schedule with 5 warmup epochs. All ProteinMD experiments are run on a single GPU. The full set of hyperparameters is summarized in Table 12.

# M. Optimization Sensitivity of Learned Key Projections

In Section 4.1 and Remark 4.2 of the main text, we describe the design trade-off between fixed key vectors ($k_j = 1$) and learned linear projections ($k_j = W^K f_j$). Fixed keys impose a purely geometric kernel, whereas learned keys increase expressivity by mixing geometry and content. In this section, we analyze the optimization sensitivity of the learned-key variant on QM9.

## M.1. Learned Keys Without Additional Normalization

To investigate the impact of learned keys, we conducted a stress test on the QM9 dataset using the standard hyperparameters defined in Appendix H, without the additional QK normalization used in our OMol25 setting. We compared the standard model (fixed keys) against a variant with learned key projections. We performed this comparison for both the full Attention mechanism and the linear Convolutional variant, training for 300 epochs across two random seeds.

The results are illustrated in Figure 4. As shown in Figure 4a, when using the full Attention mechanism, the introduction of learned keys ('use_key=True') makes training substantially more optimization-sensitive in this setting. Both runs utilizing learned keys exhibit divergence around epoch 10, with one run failing to complete. In contrast, the fixed key formulation ('use_key=False') trains smoothly.

In the linear Convolutional mode (Figure 4b), training remains stable for both configurations. However, as shown in Figure 4c, the learned keys provide no performance benefit in this QM9 setup; in fact, the model with fixed keys achieves a lower Test MAE. This suggests that for this smaller physical task, the robust geometric bias from fixed keys is preferable to the additional mixed content–geometry expressivity of learned keys.

## M.2. Mitigating Sensitivity via Regularization and QK Normalization

We further hypothesized that the optimization sensitivity in the Attention setting might be mitigated by stronger regularization. We performed a sweep of weight decay values ranging from $10^{-1}$ to $10^{-8}$ for the model with learned keys.

Figure 5 presents these results. Figure 5a shows that while high weight decay values ($10^{-1}$ to $10^{-4}$) can stabilize the training, reducing the weight decay below $10^{-4}$ immediately reintroduces the sensitivity observed in the previous experiment. Figure 5b shows that the best stable weight-decay setting still lags behind the default constant-key scenario with weight decay $10^{-8}$ (Figure 4). This indicates that regularization alone is not the only possible mitigation: in our OMol25 experiments, where learned keys are beneficial, we pair them with QK normalization to directly control query/key magnitudes before computing attention scores.

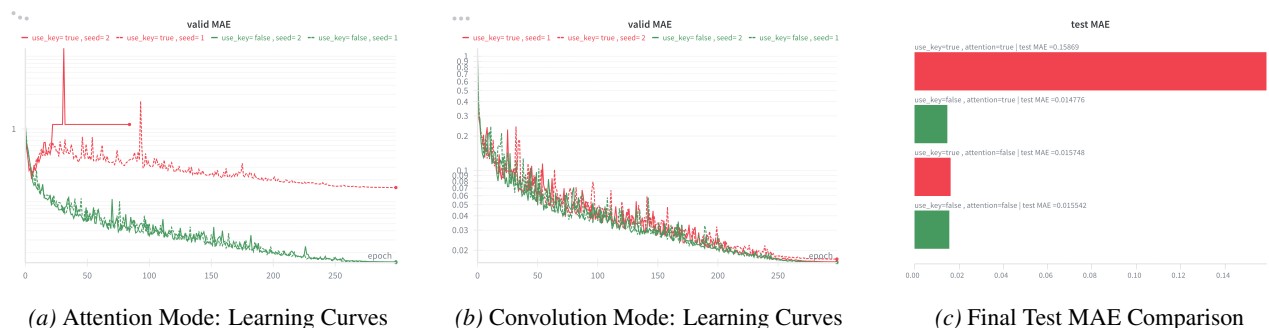

*(a)* Attention Mode: Learning Curves     *(b)* Convolution Mode: Learning Curves     *(c)* Final Test MAE Comparison

*Figure 4.* Impact of Learned Key Projections on Stability and Performance. (a) In full attention without additional normalization, learned keys are more optimization-sensitive and diverge around epoch 10. (b) In convolutional mode, training is stable for both variants, but (c) fixed keys achieve better final accuracy in this QM9 setup.

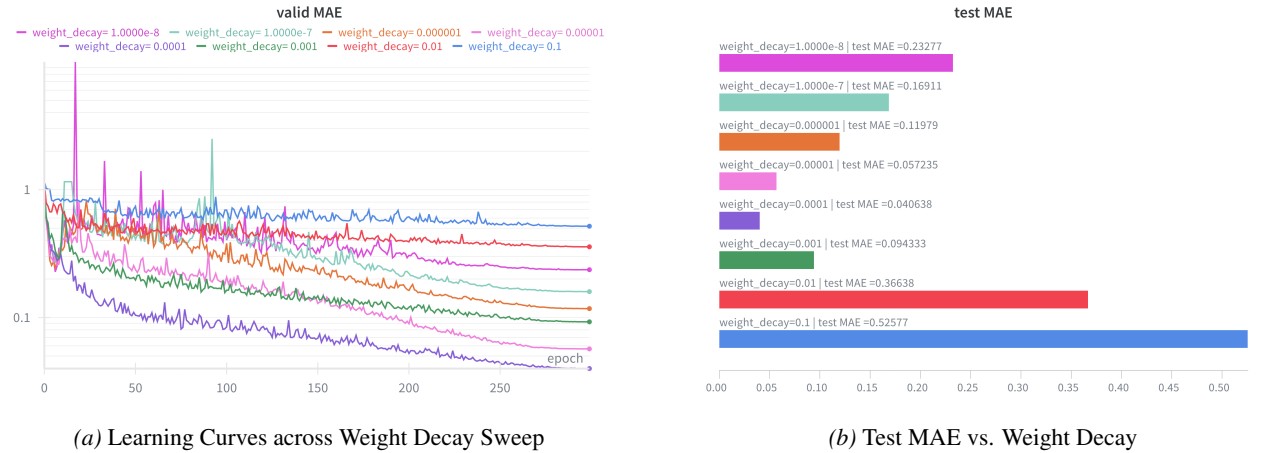

*(a)* Learning Curves across Weight Decay Sweep       *(b)* Test MAE vs. Weight Decay

*Figure 5.* Can Weight Decay Stabilize Learned Keys? (a) Strong weight decay stabilizes training, while values $< 10^{-4}$ lead to divergence. (b) The final test MAEs of each model.

**Conclusion:** These experiments confirm that for QM9-like physical tasks, fixed keys ($k = 1$) are a strong robust default rather than merely a simplification: they enforce a clean geometric kernel and train reliably under the standard hyperparameters. Learned keys remain a viable, more expressive choice when paired with stronger stabilization such as QK normalization, as used in our OMol25 experiments.

## N. Further Ablations and Analysis

### N.1. Equivariance Error

We report relative equivariance errors $|f(Rx) - f(x)|/(\frac{1}{2}(|f(x)| + |f(Rx)|))$ for Platonic transformers trained on the $\mu$ target of QM9 in Table 13. The error is the median over samples of $x$ from the validation set and $R$ from $SO(3)$. All models are approximately equivariant after training, but the larger the group is the more equivariant the models are at initialization.

*Table 13.* Equivariance error on QM9-$\mu$.

| Group | At init | After training |
|---|---|---|
| $\{\mathbf{e}\}$ | 0.21 | 0.0066 |
| Tetrahedron | 0.061 | 0.0057 |
| Octahedron | 0.028 | 0.0043 |

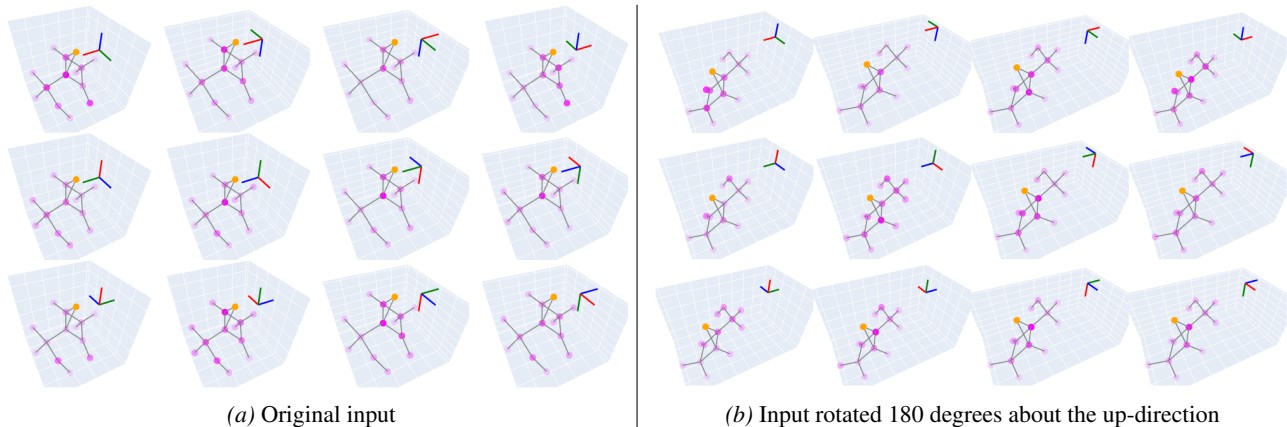

*(a)* Original input        *(b)* Input rotated 180 degrees about the up-direction

*Figure 6.* We visualize the attention score between the orange node and all others, where an increased color intensity indicates an increased attention score. The subplots correspond to 12 different frames in the same head of an octahedral Platonic Transformer layer (there are 12 more frames not visualized here). The attention is broadly focused on locality but with distinct directional biases. The equivariance of the model can be observed by comparing the attention scores in the sub-figures. For instance, the attention pattern in the top-left frame in Figure 6a is the same as the one in the top-right frame in Figure 6b, but rotated 180 degrees.

### N.2. Equivariant versus Invariant Attention Scores

We perform an ablation on equivariant versus invariant attention scores as described in Section 4.2. We train an octahedral Platonic Transformer on target $\mu$ of QM9. The model with equivariant attention scores obtains 0.01 MAE (as in Table 10) while the one with invariant attention scores reaches obtains 0.02 MAE.

### N.3. Visualizations of Learned Attention Scores

To show the directional attention learned in the attention head, we visualize examples over attention patterns in different frames $g \in \mathcal{G}$ in Figure 6.

## O. Implementing Platonic Transformers in the Fourier Domain of Finite Groups

With increasing hidden dimension (while not increasing sequence length), transformer blocks spend more and more of their total compute time in the pointwise linear layers. To improve speed it can then be worthwhile to implement the pointwise equivariant linear layers in the Fourier domain of the rotation group, a technique that has recently been successfully employed in computer vision (Bökman et al., 2025; Nordström et al., 2025). Considering the Fourier domain also sheds light on the connections between Platonic Transformers and equivariant networks with general steerable feature spaces (Cesa et al., 2022).

In this section we demonstrate how a Fourier domain implementation can improve computational efficiency in Platonic Transformers. In the Fourier domain, equivariant linear layers are block-diagonal, drastically reducing the required number of FLOPs for both forward and backward passes. We will see that with the number of hidden dimensions considered in this paper, a naive PyTorch implementation is not efficient enough to realize the reduction in FLOPs in terms of a substantial reduction in training throughput, but at a moderately higher number of hidden dimensions, there are throughput gains. This suggests that future scaling of Platonic Transformers will benefit from being implemented in the Fourier domain, and that more efficient implementations than our current one would be able to improve throughput even at smaller number of hidden dimensions.

We will use the tetrahedral symmetry group as a running example in this section. The reader is cautioned that the representations discussed in this section are representations of the rotation group, in contrast to the representations of the translation group discussed in Appendix A.

## O.1. Introduction to the Fourier Theory of Finite Groups

The representation theory of finite groups is a well studied topic with many good text books. We recommend (Serre, 1977) for more detailed background than given here. Note that we consider vector spaces over the real numbers, which leads to a slightly more involved representation theory than complex numbers, see (Serre, 1977, Section II.12).

Recall from Appendix A.1 that a representation of a group $\mathcal{G}$ is a group homomorphism $\rho : \mathcal{G} \to GL(V)$, where $V$ is a vector space. We will here consider finite real vector spaces $V = \mathbb{R}^n$ so that $\rho(g)$ can be considered real-valued invertible matrices. An irreducible representation is one where the matrices $\{\rho(g)\}_{g \in \mathcal{G}}$ can not be simultaneously block-diagonalized. Any finite group $\mathcal{G}$ has a finite number (up to ismorphisms) of irreducible representations (irreps) $\{\rho_i\}$ and they can be computed given the multiplication table of the group. Irreps are important because we can decompose any finite representation $\rho$ into a direct sum of irreps by performing a change of basis, so statements about general representations often reduce to statements about irreps.

The features in Platonic Transformers are functions from $\mathcal{G}$ to $\mathbb{R}^C$, that transform under the left regular representation as explained in Appendix B. In order words, the representation that acts on them is a direct sum of $C$ copies of the regular representation of $\mathcal{G}$. Let this representation be denoted $\tilde{\rho}$. Decomposing $\tilde{\rho}$ into irreps, we obtain

$$\tilde{\rho}(g) = Q \left( \bigoplus_i \rho_i(g)^{\oplus m_i} \right) Q^{-1} \tag{31}$$

for some multiplicities $m_i$ of each irrep and a change of basis matrix $Q$ that can be taken to be orthogonal.

Now, Schur's lemma says that any equivariant linear map between non-isomorphic irreps $\rho_i \neq \rho_j$ must be constant zero. Further, the space of equivariant linear maps between $\rho_i$ and itself is 1-, 2-, or 4-dimensional and isomorphic (as a division algebra over $\mathbb{R}$) to the real numbers, complex numbers, or quaternions depending on whether $\rho_i$ is of so-called real, complex or quaternion type. (The type of $\rho_i$ can be computed.) This means that any linear map that is equivariant from $\tilde{\rho}$ to $\tilde{\rho}$ is actually block diagonal after having performed the change of basis in (31), in particular so are the group convolutions used in Platonic Transformers.

For cyclic groups, the block-diagonalization corresponds to the fact that convolutions are pointwise multiplications in the Fourier domain[4].

## O.2. Fourier Theory of the Tetrahedral Group

Let us now consider the Tetrahedral rotation group as $\mathcal{G}$, consisting of the twelve rotational symmetries of a regular tetrahedron. This group is isomorphic to the alternating group $A_4$ and has three real irreps. The real irreps of the tetrahedral group are given by the one-dimensional trivial representation

$$\rho_1(R) = 1, \tag{32}$$

the three-dimensional standard representation

$$\rho_3(R) = R \tag{33}$$

and a two-dimensional representation $\rho_2$ that is defined as follows. Note that any element in $\mathcal{G}$ is either the identity, a rotation by $2\pi/3$ radians (there are 8 of these) or a rotation by $\pi$ radians (there are 3 of these). For the identity and rotations by $\pi$,

$$\rho_2(R) = \begin{pmatrix} 1 & 0 \\ 0 & 1 \end{pmatrix}. \tag{34}$$

The rotations by $2\pi/3$ fall into two conjugacy classes of four elements each, where one conjugacy class contains the inverses of the second. We can arbitrarily choose one of the conjugacy classes and define

$$\rho_2(R) = \begin{pmatrix} \cos(2\pi/3) & -\sin(2\pi/3) \\ \sin(2\pi/3) & \cos(2\pi/3) \end{pmatrix} \tag{35}$$

there, which implicitly defines the values for the second conjugacy class to be the inverse of the above.

---

[4]This requires working over the complex numbers, over the real numbers the pointwise multiplications turn into $2 \times 2$ matrix multiplications, again a block-diagonal structure.

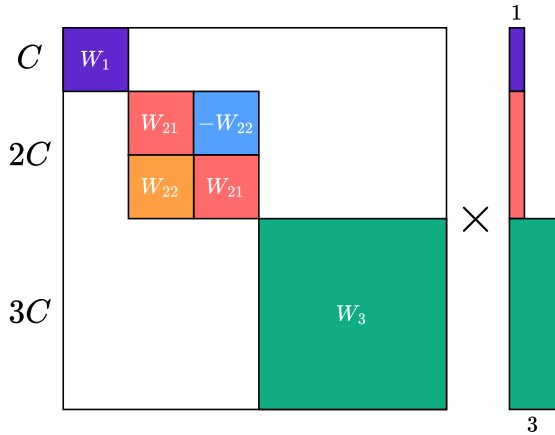

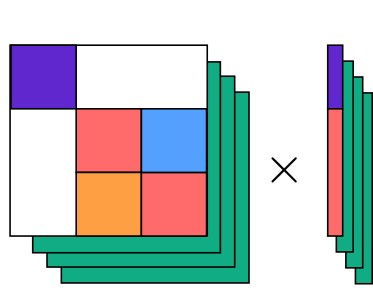

*(a)* The block-diagonal structure of an equivariant weight matrix in the Fourier domain.

*(b)* We can implement the linear layer as a batched matrix-vector multiplication with four batches.

*Figure 7.* We visualize the weight matrices for linear layers that are equivariant under the tetrahedral rotation group, implemented in the Fourier domain. Each subfigure shows weights to the left and features to the right. Purple features transform according to $\rho_1$ (or technically $\rho_1 \otimes I_C$ since there are $C$ copies of $\rho_1$), red features according to $\rho_2$ (by multiplication by $\rho_2(g) \otimes I_C$ from the left) and green features according to $\rho_3$ (by multiplication by $\rho_3(g)^\top$ from the right (if we flattened the green features, they would transform by $\rho_3(g) \otimes I_{3C}$ from the left)). The weight matrix is parameterized by the $C \times C$ matrices $W_1, W_{21}, W_{22}$ and the $3C \times 3C$ matrix $W_3$, yielding a total of $12C^2$ learnable parameters. The total number of multiplications to compute the linear layer implemented as a batched matrix-multiplication in 7b is $4 \cdot (3C)^2 = 36C^2$, yielding a $4\times$ FLOP reduction versus an ordinary layer from $12C$ to $12C$ dimensions ($144C^2$ multiplications).

It can be computed that $\rho_1$ and $\rho_3$ are both of real type, while $\rho_2$ is of complex type. Hence, equivariant linear maps from $\rho_1$ to $\rho_1$ are parameterized by one value, and the same for $\rho_3$. Equivariant linear maps from $\rho_2$ to $\rho_2$ are instead parameterized by two values (this is because $\rho_2$ splits into two irreps over the complex numbers).

It can also be computed (or recovered from general facts of the Fourier transform over finite groups) that the representation $\tilde{\rho}$ acting on features with $C$ channels in a tetrahedral Platonic Transformer splits into $C$ copies of $\rho_1$, $C$ copies of $\rho_2$ and $3C$ copies of $\rho_3$ (as a sanity check, we recover all $C + C \cdot 2 + 3C \cdot 3 = 12C$ dimensions).

As mentioned, Schur's lemma now implies that equivariant linear maps from $\tilde{\rho}$ to itself are block-diagonal. The map from copies of $\rho_1$ to copies of $\rho_1$ is parameterized by a $C \times C$ matrix, the map from copies of $\rho_2$ to copies of $\rho_2$ is parameterized by two $C \times C$ matrices (because $\rho_2$ is of complex type) and the map from copies of $\rho_3$ to copies of $\rho_3$ is parameterized by a $3C \times 3C$ matrix. Again, a sanity check gives that the full equivariant layer is then parameterized by $C^2 + 2C^2 + (3C)^2 = 12C^2$ values, which is the same as the group convolution discussed in Section 3.3.

We visualize the weight structure in Figure 7a.

**O.3. Implementation**

We implement a version of the Platonic Transformer with tetrahedral equivariance and all linear layers (i.e. in the MLP and projections in multi-head attention) in the Fourier domain. We transform back to the spatial domain at each non-linearity and at the RoPE-attention layers and to the Fourier domain after these layers. This transforming back-and-forth incurs an overhead that goes to zero as the hidden dimension increases (since it is just the $12 \times 12$ matrix $Q$ applied to each channel $C$), however it is non-negligible at low–medium number of hidden dimensions, because it involves non-contiguous reshapes.

The maximum FLOP saving that can be obtained from changing a linear layer to be in the Fourier domain is going from $(12C)^2 = 144C^2$ operations to $C^2 + (2C)^2 + 3 \cdot (3C)^2 = 32C^2$, i.e. a saving of $4.5$ times. However, in order to make the implementation more efficient in pure PyTorch, we opt to implement the mappings for $\rho_1$ and $\rho_2$ as one single $3C \times 3C$ matrix, enabling the whole linear layer to be implemented as a batched matrix multiplication with four $3C \times 3C$ weight matrices, as illustrated in Figure 7b. This batched implementation uses $4 \cdot (3C)^2 = 36C^2$ operations, yielding a maximum potential compute saving of $4$ times.

### O.4. Throughput Benchmarking

We benchmark the training time per epoch on a subset of 20k molecules on the OMol25 task, using PyTorch's `torch.compile`. These timing runs are on a single NVIDIA RTX6000 GPU. We keep all hyperparameters constant as in the main experiments, except for varying the number of hidden dimensions. The results are presented in Table 14. It is clear that as we increase the number of hidden dimensions, a Fourier implementation starts paying off more and more. Notably, since the standard spatial implementation is equal to non-equivariant Transformers in computational cost, the efficiency improvement of the Fourier implementation is a benefit of equivariant architectures over non-equivariant ones. We emphasize that our Fourier implementation is not well-optimized, so further throughput improvements should be available.

*Table 14.* Training times per epoch (seconds) on a subset of OMol25 with 20k examples. We compare a tetrahedral Platonic Transformer implemented in the spatial domain with one implemented in the Fourier domain.

| | **Hidden dimension** | | | | | |
|---|---|---|---|---|---|---|
| **Implementation** | 576 | 864 | 1152 | 1440 | 1728 | 2016 |
| Spatial (standard) | 18 | 23 | 29 | 40 | 49 | 63 |
| Fourier | 19 | 22 | 27 | 32 | 38 | 45 |

