# OpenReview forum: "Platonic Transformers: A Solid Choice For Equivariance"
_ICML.cc/2026/Conference — ICML 2026 regular_

### Official Review · Reviewer_5ncm · 2026-03-04

**Soundness:** 3
**Presentation:** 3
**Significance:** 3
**Originality:** 3
**Overall Recommendation:** 5
**Confidence:** 3

**Summary:**

The paper introduces the Platonic Transformer, a Transformer framework designed to incorporate equivariance to continuous translations and discrete roto-reflection symmetries by representing features relative to a finite set of reference frames derived from Platonic solid symmetry groups. The main idea is to associate each reference frame with a structured attention branch and enforce equivariance in the linear layers through principled weight sharing, effectively implementing group-convolution-style maps while preserving the standard Transformer computation graph. A further contribution is the observation that RoPE-based attention admits a dynamic convolution interpretation, which also motivates a linear-time variant when softmax is removed.
The paper evaluates the approach across a broad range of tasks, including CIFAR-10, ScanObjectNN, ProteinMD, QM9 generation, and OMol25, and shows that this structured geometric bias can improve over trivial-group baselines while remaining compute-aligned with standard Transformer architectures under matched effective dimensionality. Overall, I find this to be a conceptually elegant and technically meaningful contribution: it offers a clean way to inject symmetry structure into Transformers without redesigning the entire architecture, and the empirical scope is broader than is typical for papers in this area.

**Compliance With Llm Reviewing Policy:**

Affirmed.

**Ethical Review Concerns:**

This paper offers an elegant and technically meaningful way to inject geometric inductive bias into Transformers through discrete Platonic symmetry groups while largely preserving the standard Transformer computation graph. The work is strong in originality (clean group-structured weight sharing), clarity (the design is easy to isolate), and breadth (multiple domains). My main initial concerns were (i) the lack of a direct quantitative study of approximate equivariance under continuous rotations, (ii) mixed results on ScanObjectNN where “Flop” outperformed tetrahedral, and (iii) ambiguity in the OMol25 takeaway (budget efficiency vs scaling superiority). The rebuttal substantially addresses these points. I therefore update my recommendation to 5/6 (Accept), with increased confidence compared to my initial assessment.

**Final Justification:**

The rebuttal substantially strengthens the paper and resolves the main concerns raised in my review. I therefore update my recommendation to 5/6 (Accept), with increased confidence compared to my initial assessment.

**Key Questions For Authors:**

1. In Table 2, why does the “Flop” baseline outperform the tetrahedral variant on ScanObjectNN? Could this indicate that stronger Platonic symmetry can over-regularize some datasets?

2. For OMol25, should the main takeaway be budget efficiency, eventual scaling potential, or both? It would help if the claim were framed more explicitly.

**Limitations:**

Yes.

**Strengths And Weaknesses:**

Strengths
1. Elegant and appealing core idea.
The paper introduces geometric inductive bias through structured weight sharing over discrete reference frames, rather than through a heavily modified attention mechanism. This makes the method conceptually clean and attractive as a drop-in architectural alternative.
2. Strong theoretical framing.
The paper provides a principled end-to-end equivariance result in the discrete-group setting and offers an insightful interpretation of RoPE through the lens of dynamic group convolution. This substantially strengthens the conceptual depth of the work.
3. Broad empirical validation.
The evaluation spans multiple domains (2D vision, 3D point clouds, molecular dynamics, and molecular generation), which gives the paper wider relevance than many specialized equivariant Transformer works. The CIFAR-10 and ProteinMD results are particularly supportive of the proposed inductive bias.
4. Thoughtful compute-budget framing.
The paper is commendably careful in comparing models under matched effective capacity, and the appendix explicitly acknowledges the trade-off between group size and per-frame feature dimensionality. This makes the empirical comparisons more credible.

Weaknesses
1. Approximate rather than continuous equivariance.
While the model is exactly equivariant to the chosen finite group $G$, its relationship to continuous $SE(3)/E(3)$ symmetry is only approximate. The paper would be significantly stronger if it included a direct quantitative study of equivariance error under random continuous rotations and how this error changes with the choice of group.
2. Some empirical results are mixed.
The method is broadly competitive, but not uniformly dominant. For example, on ScanObjectNN, the “Flop” baseline outperforms the tetrahedral variant. This suggests that Platonic symmetry is not always the most effective inductive bias and that task dependence should be discussed more explicitly.
3. OMol25 evidence mainly supports budget efficiency, not yet full scaling superiority.
The 120-hour comparison is useful and supports the claim that the approach is efficient under constrained compute. However, the stronger literature-reported baselines still outperform the reproduced runs, so the current evidence supports efficient competitiveness more than definitive superiority at scale.

---

> ### Author Rebuttal · Authors · 2026-03-31
>
> Thank you for your detailed and thoughtful review. We are particularly happy to see the description of the method as *a conceptually elegant and technically meaningful contribution*.
>
>
> In the following, we wish to address the weaknesses and questions asked.
>
> **W1. Approximate equivariance.**
>
> Please refer to the answer to Reviewer uEBM for an evaluation of equivariance error under the full rotation group.
>
> **W2/Q1. Why does the “Flop” baseline outperform the tetrahedral variant on ScanObjectNN?**
>
> The reviewer's intuition is indeed right, we would argue this is a case of over-regularization. As noted in Section 5.1, objects in ScanObjectNN are aligned with a canonical up-direction. Because the data does not exhibit full 3D rotational symmetry, enforcing strict Tetrahedral equivariance forces the model to share weights across orientations (e.g., upside-down) that do not exist in the dataset, unnecessarily restricting the kernels, or unnecessarily lowering the number of effective parameters. The "Flop" baseline outperforms the others because it perfectly matches the true underlying symmetry of the dataset: everyday objects typically exhibit bilateral (left/right) symmetry, making it the optimal inductive bias for this specific task. Furthermore, using an overly restricted model enables an increase in the hidden dimension without sacrificing compute time, resulting in improved performance (see the ImageNet-1K table provided to Reviewer MjAW). To make this clearer in the camera-ready revision, we will elaborate on this in the experiments section and add a corresponding ScannetNN ablation study. This will better illustrate how equivariance can be utilized in tasks lacking equivariance to the entire group of rotations. We will add the interesting discussion in our revised manuscript.
>
> **W3/Q2. For OMol25, should the main takeaway be budget efficiency, eventual scaling potential, or both?**
>
> We believe the main takeaway should explicitly be both budget efficiency and eventual scaling potential, and we thank the reviewer for pointing out that this needs to be framed more clearly.
>
> Empirically, the current OMol25 result already goes beyond a strictly small-scale proof of concept as it is run on the 34.3M-sample Neutral split under a fixed 5-day / 4-GPUs(H100) budget. In that matched-compute setting, Platonic improves over the reproduced eSEN baseline on energy by approximately 1.7x, while remaining highly competitive on forces. So we believe the current table shows not just that the method works under a constrained budget, but already hints that it remains effective in a fairly large-scale regime.
>
> At the same time, we would like to note that the scaling argument is also architectural. Our method preserves the standard Transformer computation graph and overall computational footprint, so the geometric inductive bias is introduced, for example, without moving to heavier machinery like a tensor field. Thus, it inherits the favorable hardware efficiency and wall-clock scaling behavior of the standard transformer as the model scales.
>
> We would also like to inform you that we are running more experiments on the OMol dataset with a bigger compute budget. The learning curves so far are consistent with what we see across the other benchmarks: improved optimization, faster convergence, and expected stronger final performance on this equivariant task. In the revision, we will make three points more explicit. First, that the current OMol25 result already demonstrates competitiveness at scale; second, that the model’s scaling potential also comes from preserving the standard Transformer computation graph and footprint; and third, that on equivariant tasks such as OMol25, the tetrahedral variant shows better optimization and generalization than the trivial variant while remaining compatible with standard Transformer-style scaling in light of the existing and ongoing new experiments.
>
> We thank you again for your insightful comments and would appreciate you considering raising your score if you believe they have been addressed sufficiently, or follow up with any questions we may help clarify further.

---

> > ### Author Rebuttal · Reviewer_5ncm · 2026-04-01
> >
> > The rebuttal substantially strengthens the paper and resolves the main concerns raised in my review. I therefore update my recommendation to 5/6 (Accept), with increased confidence compared to my initial assessment.

---

> > > ### Author Response · Authors · 2026-04-07
> > >
> > > Dear Reviewer, thank you for your time, constructive feedback, and careful review of our rebuttal. We are very glad that our responses and revisions were able to fully resolve your concerns, and we deeply appreciate your updated score and recommendation. We will ensure all the discussed improvements are carefully incorporated into the final manuscript.

---

### Official Review · Reviewer_iwnT · 2026-03-12

**Soundness:** 2
**Presentation:** 3
**Significance:** 2
**Originality:** 2
**Overall Recommendation:** 2
**Confidence:** 4

**Summary:**

This paper proposes a framework for achieving equivariance to continuous translations and discrete roto-reflections in Transformers. The core idea is to prepare the relative positional encoding, RoPE, relative to reference frames defined by discretized rotation groups of Platonic solids, where each reference frame corresponds to an attention head. The paper also designed the equivariant linear layers, invariant and equivariant attention score calculation, and linear RoPE attention with constant key vectors. Experiments are conducted on CIFAR-10, ScanObjectNN, QM9, ProteinMD, and OMol25.

**Compliance With Llm Reviewing Policy:**

Affirmed.

**Final Justification:**

I maintain my original evaluation of this work: it has recognizable but marginal novelty, and the experimental evidence shows that it achieves overall comparable performance as the baselines, but does not show significant added value.

I agree with the authors that the network parameters can be significantly reduced without sacrificing performance, but the number of parameters is not a primary concern about the efficiency of a model. FLOPs and wall-clock time at inference are more important efficiency metrics.

The reduced number of model parameters has always been a key advantage of almost all equivariant models, but it hasn't convinced the broader community to unanimously adopt equivariant models all these years. Furthermore, to respond to the authors' claim in the rebuttal, "the key point is that the observed gains come from the proposed equivariant weight-sharing itself, not from target-by-target tuning or a heavier specialized architecture." On the one hand, equivariant weight-sharing has been widely studied, which is not a conceptual contribution of this paper. On the other hand, it is hard to judge whether the empirical gain of equivariant inductive bias, which is already thin in the reported results to start with, will diminish when more data or computing power are available.

Successful equivariant models in the literature actually show significantly better performance or robustness that baseline methods couldn't match at comparable compute, or superior efficiency at comparable performance, at least in certain applications. I actually think this paper has the potential to achieve this. For example, the authors may run a 3D point cloud-related task where equivariance and robustness to orientations are critical (as opposed to the reported non-equivariant ScanObjectNN task) to more convincingly show its superiority.

Therefore, I keep my rating of this paper as "Reject", but I believe that an updated version with a better-designed and presented experimental section will make this paper much stronger.

**Key Questions For Authors:**

Please address each of the weaknesses I listed above.

**Limitations:**

This paper did not discuss the limitations.

**Strengths And Weaknesses:**

Strengths:

1. The paper is clearly written, and the connection between RoPE, reference frames, and group equivariance is presented in a clean way.

2. The overall design is simple and easy to implement.

3. The framework is evaluated across a diverse set of benchmarks spanning 2D vision, 3D point clouds, and molecular tasks.

Weaknesses:

1. The novelty claim is overstated. The core ingredients, including equivariant Transformers, finite group discretization of SO(3), frame-based processing, and RoPE, are all well-established. The specific contribution is combining them such that the group axis maps onto attention heads. This mapping is fairly mechanical: it is essentially the same observation that makes group convolutions a natural generalization of regular convolutions.

2. The "zero additional cost" framing is misleading. The paper claims equivariance is achieved at no additional computational cost. However, the model achieves this by keeping the total channel dimension fixed while reducing the number of channels per orientation by a factor of |G|. This is a trade-off: the model has strictly fewer channels per frame than a non-equivariant baseline with the same parameter count. The paper does not adequately acknowledge this trade-off or characterize when it is favorable.

3. The equivariant vs. invariant attention design choice is underanalyzed. Section 4.2 distinguishes between equivariant and invariant attention scores, presenting the former as a more expressive formulation. However, there is no systematic empirical analysis showing when this distinction matters.

4. The linear-time convolutional variant lacks a performance-efficiency analysis. The constant-key formulation and its linear complexity are theoretically well-motivated, but the paper does not present a systematic analysis of the accuracy-efficiency trade-off.

5. Empirical results are competitive but not convincing. The OMol25 comparison to eSEN is conducted under a constrained compute budget, with the authors acknowledging the literature benchmarks used substantially more compute. Results on QM9 are reported with a single fixed set of hyperparameters across all targets, which the authors justify as a deliberate choice but which makes the comparison to target-tuned baselines difficult to interpret. The overall experimental story supports "competitive at matched compute" but does not provide a more convincing support for the advantages of the proposed method.

6. This paper did not discuss its limitations.

---

> ### Author Rebuttal · Authors · 2026-03-31
>
> We thank the reviewer for highlighting our writing and ideas as *clearly written*, *clean*, and *simple and easy to implement*, and for pointing out potential weaknesses we address below.
>
> **Novelty claim.** The reviewer is right that the equivariant RoPE construction is similar to a group convolution. In hindsight, the construction is very natural, which is a strength of the method, not a weakness. Prior works on equivariant transformers have not identified this natural RoPE construction and focus on computationally expensive approaches. The novelty claim is not just the technical innovation but actually showing that equivariance does not have to come at the cost of compute, which seems to be the general belief. This paper is the first to show this in precisely controlled experiments (flops/operations are exactly the same across different choices for symmetries)
>
>
> **Zero additional cost.** We compare networks with identical computational demands and equal *total* channels ($|G|\cdot C$). The **zero additional cost** refers to no extra computational costs in this setting. Here, learnable parameters decrease with $|G|$, yet our experiments show that the equivariant networks perform well despite this reduction in parameters. So not only is it zero cost, but it also reduces the actual parameter count, hence it has a small memory footprint.  In addition, we present experiments with different groups and different parameters (Tab 1 rev. MjAW) to provide insights into the tradeoff which will be added to the camera-ready version as well.
>
> **Invariant attention.**
> We agree that this is an interesting comparison; we ablated equivariant vs. invariant attention scores on the QM9 $\mu$-target using an octahedral Platonic Transformer. The equivariant version achieves 0.010 MAE (Table 8), while the invariant version obtains 0.019 MAE. Notably, both still outperform the non-equivariant trivial baseline (0.028 MAE).
>
> **Convolution variant.**
> Let us first summarize the presented results from the paper. We have included convolutional variants in Tables 1 (CIFAR-10), 2 (ScanObject-NN), and 8 (QM9 regression). We have also provided Figure 3 showing how the time of the forward pass scales with sequence length. The results reported in the Tables show that the performance of the convolutional variants compared to the attention variants varies across different tasks. For CIFAR10 and ScanObjectNN, the attentional variants clearly outperform the convolutional, whereas on QM9, it varies depending on the target. Regarding the time complexity tradeoff, it depends both on the sequence length and embedding dimension of the model, on how much a linear attention implementation improves the throughput of the model. To supplement Figure 3, we have computed average training times in minutes per epoch for Platonic Transformers trained on QM9 on an RTX6000 GPU in the table below.
>
> | | Attention | Convolution |
> |---|---|---|
> | Trivial      | 2.8 | 2.1   |
> | Tetrahedron     | 2.4 | 1.8   |
> | Octahedron      | 2.4| 1.8   |
>
> **Empirical results.**
> The current experiments are best read as controlled comparisons, not as an attempt to maximize every benchmark through target-specific tuning. The main goal is to isolate the effect of the equivariant constraint by comparing constrained and unconstrained models in a standardized, compute-aligned setting. This is exactly why, on QM9, we use a single fixed hyperparameter configuration across all targets, even though several baselines tune hyperparameters separately for each target. That choice likely puts us at a disadvantage in absolute numbers, but it makes the comparison between the trivial and equivariant variants much cleaner and more informative.
>
> Even under this deliberately conservative protocol, the comparison remains informative. On OMol25, we evaluate under a matched compute budget. More broadly across all experiments (please also refer to our response to Reviewer *MjAW*, where we provide new empirical results on ImageNet), the key point is that the observed gains come from the proposed equivariant weight-sharing itself, not from target-by-target tuning or a heavier specialized architecture, while still keeping the scalability and hardware efficiency of a standard Transformer.
>
> **Limitations**
> We will add a limitations section acknowledging our model provides exact equivariance only for discrete $O(n)$ subgroups, approximating full $O(n)$-equivariance (see response to Reviewer uEBM for equivariance error quantification). We will also clarify our evaluation boundaries: prioritizing fair comparisons via matched compute and fixed hyperparameters may leave absolute performance gains on the table. Finally, while budget-efficient, the model's scaling laws remain to be fully characterized.
>
> We thank you for your constructive feedback and would appreciate you considering raising your score if you believe they have been addressed sufficiently, or follow up with any questions we may help clarify further.

---

> > ### Author Rebuttal · Reviewer_iwnT · 2026-04-04
> >
> > I thank the authors for their detailed rebuttal. The response and additional experiments partially addressed my concerns.
> >
> > I maintain my opinion that the novelty of the proposed method is recognizable but marginal. I would be more forgiving of the "novelty" if the empirical performance were strong, but I found the experimental evidence also mixed, and the performance improvements across various tasks are marginal.
> >
> > I see that in response to reviewer MjAW, you provided an efficiency comparison using different polyhedron groups, but their performance is not shown. Is there a trade-off when selecting what polyhedron to use? Table 1 in the paper shows the impact of group selection, but only on the CIFAR-10 image benchmark. An analysis on 3D point cloud tasks would be more relevant.
> >
> > Also in your response to MjAW, flop_2d_2 actually performs worse than the baseline, showing that the cut in effective feature dimension per group element could hurt performance. Although you show that using a wider network achieves better results with similar training efficiency as the baseline, I believe test-time efficiency is more important, and I believe the flop_2d_2 (non-wide) is more comparable to the baseline in terms of test-time efficiency.
> >
> > Taking a closer look, I think the experiment section could be significantly improved to make the messages clearer.
> >
> > For example, what is Flop in Tables 1 and 2? In the main text 5.2 ScanObjectNN paragraph you say the comparison is between "trivial vs tetrahedron vs horizontal flips", which makes me think that the "Flop" in table 2 should be "Flip". And the term is never referred to or explained in the main text.
> >
> > The main text analysis in Section 5.2 non-equivariant tasks is also ambiguous and does not give an effective interpretation or conclusion from the corresponding tables 1 and 2. For example, for the CiFAR10 experiment, "The comparison between the full attention and linear-convolutional shows a significant impact of attention over the linear complexity dynamic convolution counterpart." For the ScanObjectNN experiment, "The results, shown in Table 2, again highlight the impact of equivariance and weight sharing." These sentences are neutral and does not inform readers what design is better. "While the
> > quadratic-cost attention mechanism offers greater expressive power, the linear-time convolutional variant provides a significant speed-up". But Table 2 does not show any speed-related information.
> >
> > Given above, I maintain my current score.

---

> > > ### Author Response · Authors · 2026-04-07
> > >
> > > Dear Reviewer, we thank you again for your continued engagement.
> > >
> > > **On Novelty**
> > >
> > > You noted that the novelty is "recognizable but marginal" and that you would be more forgiving if the empirical performance were strong. We hear that. Rather than re-arguing novelty, we focus on the empirical case below.
> > >
> > > **On the experimental evidence (re: "empirical results competitive but not convincing")**
> > >
> > > We think this concern partly stems from reading the experiments as benchmark attempts. Below we would like to clarify what they are designed to show. They are controlled comparisons supporting two concrete claims:
> > >
> > > *Claim 1: Equivariance at zero computational cost, vastly reduced parameter counts, no loss in performance.* Across every benchmark (CIFAR-10, ScanObjectNN, QM9, ProteinMD, OMol25, ImageNet-1K), the Platonic Transformer matches or outperforms the trivial baseline at identical FLOPs while using dramatically fewer parameters (the reduction equals $|G|$: 2x for flop, 4x for cyclic_4, 12x for tetrahedron, 24x for octahedron, verifiable from parameter counts in the paper). As a new rebuttal-phase result on ImageNet-1K (see our response to Reviewer MjAW), `flop_2d_2` nearly matches the trivial ViT (79.17% vs 79.53%) at half the parameters; when scaled to match the baseline footprint, it actually improves (80.41%) without increasing training time. This is a consistent pattern across six tasks spanning 2D vision, 3D point clouds, and molecular domains.
> > >
> > > *Claim 2: Equivariance acts as a beneficial inductive bias, including on tasks without strict equivariance.* On molecular tasks (QM9, ProteinMD, OMol25), equivariant variants consistently improve over the trivial baseline in final performance and convergence speed. On CIFAR-10 and ImageNet-1K, where the data has no rotational symmetry, equivariant weight-sharing still acts as an effective regularizer. On ScanObjectNN, matching the bias to the data's true symmetry (bilateral flip rather than full rotation) yields the best result, showing that flexibility in choosing the symmetry group is a practical strength.
> > >
> > > As for OMol25, we will more clearly frame it in the revision as a proof of concept that these benefits extend to large-scale settings (matched compute parity, 34.3M samples, ~1.7x energy improvement over reproduced eSEN), while acknowledging that a full scaling analysis remains future work.
> > >
> > > As we can see, across six tasks, equivariant weight-sharing consistently matches or improves performance at identical FLOPs and up to 24x fewer parameters. Would you agree that this constitutes convincing evidence for the two claims above?
> > >
> > > **Re: polyhedron trade-offs on 3D tasks**
> > >
> > > This analysis exists in the paper. Table 8 reports QM9 results for both Octa and Tetra. Performance is task-dependent: Octa gives lower errors on energetic properties (HOMO, LUMO, gap), Tetra on spatial/charge properties ($R^2$, alpha, mu), illustrating the trade-off between stronger symmetry bias and reduced per-frame feature capacity.
> > >
> > > **Re: flop_2d_2 and test-time efficiency**
> > >
> > > We agree the non-wide `flop_2d_2` is the fairer test-time comparison. This result supports our claims: a sub-0.5% accuracy gap while halving parameters, with identical inference speed. At matched test-time cost, equivariance loses almost nothing while providing substantial memory savings. The wide variant shows that reinvesting freed capacity improves beyond the baseline (80.41% vs 79.53%).
> > >
> > > **Terminology and Section 5.2**
> > >
> > > "Flop" follows the terminology of Bokman et al. We agree it was not introduced in the main text and will fix this. For Section 5.2, the speed comparison is already in Figure 3; we will make the text draw explicit conclusions: attention offers peak performance (outperforming convolution on CIFAR-10 and ScanObjectNN), while the linear-time variant trades some accuracy for speed (1.8 vs 2.4 min/epoch on QM9, Tetra setting).
> > >
> > > We appreciate that your feedback has pushed us to sharpen our framing and presentation. We believe we are in agreement on several points: you noted the method is simple and easy to implement, you acknowledged the results are competitive at matched compute, and you rightly pointed out that the presentation needed to be more precise. We have committed to concrete revisions on all of these.
> > >
> > > Where we hope to find further agreement: do you agree that (1) matched or improved performance at up to 24x fewer parameters and identical FLOPs is a meaningful practical contribution, and (2) the consistency of this across six diverse tasks supports equivariant weight-sharing as a broadly useful inductive bias? We are not asking you to accept scaling claims we have not yet demonstrated, and we will more explicitly mark the scope of our experiments in the revision. We only ask whether these controlled comparisons provide sufficient evidence for the properties we claim.
> > >
> > > We would value your perspective on this.

---

### Official Review · Reviewer_MjAW · 2026-03-13

**Soundness:** 4
**Presentation:** 4
**Significance:** 2
**Originality:** 3
**Overall Recommendation:** 5
**Confidence:** 4

**Summary:**

This paper studies expressive and efficient equivariant learning and proposes a weight sharing strategy for conventional Transformers that constrains them to produce equivariance representations. The main contribution is the resulting Platonic transformer. Experiments on non-equivariant tasks show comparable accuracy to strong baselines. Experiments on equivariant tasks (e.g., protein/molecule datasets) show a marginal improvement in accuracy and strong improvement in stability.

**Compliance With Llm Reviewing Policy:**

Affirmed.

**Final Justification:**

My initial rating was positive, and the author response across reviewers has been convincing.

**Key Questions For Authors:**

N/A

**Limitations:**

The authors state that there may be negative societal impacts, but do not list any.

**Strengths And Weaknesses:**

**Soundness**. The submission is technically sound and the claims of equivariance is theoretically supported. The methods are appropriate; it would, however, be nice to see the utility of the equivariant constraint play out in the empirical analysis (e.g., lower training times, higher sample efficiency, better generalization)

**Presentation**. The submission is clearly written and well structured. A very pleasant read.

**Significance**. The paper addresses a challenging issue, but the experiments show limited impact on predictive performance.

**Originality**. The work appears novel.

---

> ### Author Rebuttal · Authors · 2026-03-31
>
> We thank the reviewer for their time in giving feedback on our work. We appreciate that you recognize our work as *technically sound* and our claims *theoretically supported*. We are also pleased that you found the paper to be *well structured and novel*.
>
> **Regarding the utility of equivariant constraint:**
> We appreciate the reviewer's comment regarding the utility of the equivariant constraint. We respectfully argue that the manuscript already highlights these benefits: in our experiments, the proposed constraint consistently either outperforms the trivial baseline or matches its performance while requiring significantly fewer parameters and less compute time. Therefore, even when applied to non-equivariant problems, the constraint provides distinct computational advantages.
>
> To strengthen this argument, we have provided additional experiments:
>
> We applied our method to ImageNet-1K, a large-scale, non-aligned dataset. While not heavily optimized for State-of-the-Art performance, the results demonstrate that our `flop_2d_2` model achieves parity with the trivial ViT baseline while halving the parameter count. This efficiency gain allows for two distinct deployment strategies: prioritizing efficient training, or scaling the model capacity (e.g., the `wide` configuration) to match the baseline's computational footprint, which results in a clear performance improvement.
>
> Imagenet-1K:
> | Model | hidden_dim | Params | Best Top-1 | Epoch Time |
> | :--- | :--- | :--- | :--- | :--- |
> | trivial_2 | 768 | 78.9M | 79.53% | 9.4 min |
> | flop_2d_2 | 768 | 39.9M | 79.17% | 7.6 min |
> | flop_2d_2 (wide) | 1088 | 79.5M | 80.41% | 9.4 min |
>
> We also measure the forward, backward, and optimization step times for a Platonic Transformer on 3D point cloud input. Our experiment shows that the equivariant version has faster training time due to fewer actual parameters and the time required to update them. Following is the comparison summary in mean ms per step,
>
> | Solid | Forward | Backward | Optimizer Step   |   Total |
> | :----- | :----- | :----- | :----- | :----- |
> | **trivial** | 143.95 (1.02x) | 245.71 (0.97x) | 38.01 (8.46x) | 428.06 (1.07x) |
> | **tetrahedron** | 142.55 (1.01x) | 254.14 (1.00x) | 5.95 (1.32x) | 402.97 (1.01x) |
> | **octahedron** | 142.00 (1.00x) | 253.88 (1.00x) | 4.60 (1.00x) | 400.79 (1.00x) |
> | **icosahedron** | 141.54 (1.00x) | 253.46 (1.00x) | 4.50 (1.00x) | 399.81(1.00x) |
>
>
> **Regarding impact on predictive performance**
> We believe that the impact of a scalable linear time equivariant transformer to be methodological as well as through empirical results. The performance of different equivariant transformers can be considered comparable, although our approach shows ease of scalability (Fig 3, page 7) as well as significant improvement in Energy computation for OMol25 (Tab 5, page 8).
>
> **Regarding negative societal impacts:**
> We followed the ICML guidelines for foundational research where societal implications are well-established. To be more explicit as requested: the primary potential negative impacts are those common to advanced generative modeling and scientific ML, namely the risk of dual-use (e.g., repurposing molecular generation for harmful compounds) and the environmental cost of large-scale training on datasets like OMol25. We will update the Impact Statement to explicitly name these established considerations.
>
> We will incorporate the new ImageNet-1k experiment and training time details, as well as update the societal impact paragraph in the revised manuscript.

---

> > ### Author Rebuttal · Reviewer_MjAW · 2026-04-03
> >
> > Thank you for your response. I will maintain my score.

---

> > > ### Author Response · Authors · 2026-04-07
> > >
> > > We thank the reviewer for their time and for confirming that our rebuttal adequately addressed their concerns.

---

### Official Review · Reviewer_uEBM · 2026-03-13

**Soundness:** 3
**Presentation:** 3
**Significance:** 3
**Originality:** 3
**Overall Recommendation:** 4
**Confidence:** 4

**Summary:**

This work proposes a scalable alternative to pre-existing rotational equivariant architectures that achieves equivariance in discrete subgroups of rotations. The proposed transformer, referred to as the Platonic transformer, performs an attention operation by lifting the input to a discrete rotational subgroup by describing it in a reference frame defined for each group element. After the lift to the group, the authors propose a per-head/reference-frame computation of a rotary positional encoding (RoPE) used in the attention operation, which ensures the preservation of the equivariance property. In addition to the proposed equivariant attention operation, the linear layers and the corresponding MLP following the attention operation are replaced with equivalent equivariant  layers. Although the proposed design is only equivariant to a discrete group due to the requirement of lifting the input to the group of interest, the authors claim that this tradeoff allows them to retain the transformer’s computational graph and inherit its flexibility and speed. In addition to the model design, the authors provide an analysis showcasing how their formulation in the case of linear attention corresponds to input-dependent group convolutions. The authors provide an extensive evaluation of the proposed architecture across a wide range of tasks, demonstrating its flexibility and the ability to achieve performance comparable to that of more expensive, specialized equivariant architectures.

**Compliance With Llm Reviewing Policy:**

Affirmed.

**Final Justification:**

The authors' rebuttal addressed most of my questions regarding the paper.
The only main concern that remains partially unresolved is the claim about the method scalability. I understand that the computational constraints can limit the ability to showcase an extensive scaling ablation. Nevertheless the current claim of improved scalability compared to more specialized equivariant networks is based mainly on the similarity of the proposed architecture to the transformers architecture. The comparable performance in the OMol25 benchmark provides a single data point, and is not sufficient to indicate a scaling trend. As such it is difficult to evaluate the correctness of the scalability claim fully, until a scaling analysis is performed. Due to this limitation I keep my recommendation of weak accept.

**Key Questions For Authors:**

- How does the achieved performance of the Platonic transformer architecture scale as the computational resources increase? Is there a point at which it can outperform the specialized equivariant alternatives
- What is the equivariant error introduced by approximating the whole group of rotations by only a discrete subgroup?

**Limitations:**

Yes

**Strengths And Weaknesses:**

Strengths:
- The proposed architecture uses widely available pre-existing modules, making it easy to implement without requiring specialized computational packages for equivariant operations and models. This makes it easy to be adopted by a wider audience.
-  The experimental evaluation showcases that although the proposed architecture is only approximately equivariant, the effects of this approximation on the performance are not significant. As a result, it is a good choice for applications that value flexibility and ease of implementation.
-  The additional analysis and the reduction of the linear attention layer to a group convolution provide interesting insights into the equivalence between different equivariant methods.

Weaknesses:
- There is limited discussion of the equivariance error introduced by approximating the rotation group with only a finite subgroup. A more thorough analysis of how much error is introduced by this approximation will give a more valuable insight into the tradeoff between exact equivariance and the flexibility and speed of the proposed Platonic transformer.
- Some of the experimental results seem to indicate that the method underperforms existing equivariant architectures. While the authors discuss the limitation in computational resources, there is an implicit assumption that with additional resources, the Platonic transformer will be able to scale better and outperform the equivariant alternatives. Although this assumption is reasonable, it is not supported by any analysis of the scaling laws of the proposed transformer.

---

> ### Author Rebuttal · Authors · 2026-03-31
>
> We thank the reviewer for their constructive feedback.  We are glad the reviewer recognizes that the approximate equivariance has a negligible impact on performance, and finds our reduction to group convolution to be an *insightful contribution* connecting different equivariant frameworks.
>
>
> **W1 and Q2: Discussion on equivariance error.**
>
> To see the effect of discretizing the group, we compute equivariance errors for three settings as shown below.
> We have evaluated the relative equivariance error $| f(x) - f(R x) | / (\frac{1}{2}(|f(x)|+|f(R x)|))$ where $R$ is sampled from $SO(3)$ and $x$ is sampled from the validation set of QM9. We report the median error for Platonic Transformers $f$ trained on the $\mu$ target of QM9, equivariant under the symmetries of different solids in the table below. It is clear that at initialization, the tetrahedral and octahedral models are much more equivariant than the trivial (non-equivariant) baseline. After training, all models are approximately equivariant.
>
> |                 | At init         |   After training |
> |-----------------|-------------|--------------------|
> | Trivial         | 0.21    | 0.0066         |
> | Tetrahedron     | 0.061   | 0.0057         |
> | Octahedron      | 0.028   | 0.0043         |
>
>
>
>
> **W2 and Q1: Scaling of our Platonic transformer.**
>
> We thank the reviewers for raising this important question regarding the scaling potential of the Platonic Transformer. While providing an exhaustive scaling ablation is beyond our current computational budget, we strongly expect that as computational resources increase, the model will not only scale highly favorably but also maintain and likely increase its advantage over specialized equivariant alternatives. We note the following facts in relation to our expectation,
>
> First, while we cannot yet make definitive claims about ultra-large-scale performance, our current empirical observations point toward a very promising scaling trajectory. The OMol25 results in our draft offer a practical, if preliminary, proxy for this behavior. They demonstrate that the Platonic Transformer reduces training energy by approximately 1.7x while remaining highly competitive on forces, suggesting the architecture remains quite viable and efficient as we move toward larger regimes. Furthermore, this efficiency seems to be part of a broader pattern. Across our existing benchmarks, and in the early results from our ongoing trivial vs tetrahedral ablations, we generally observe that the model's equivariant constraints encourage better optimization, faster convergence, and stronger final performance. Because the model achieves these generalization benefits without inflating the underlying computational footprint, we have good reason to expect that scaling up will naturally amplify its performance advantage over trivial variants.
>
> Beyond these empirical observations, our scaling argument is fundamentally architectural. The defining bottleneck for most specialized equivariant networks, such as those that rely on tensor fields, higher-order group convolutions, or Clebsch-Gordan tensor products, is the significant computational overhead they inevitably introduce. The Platonic Transformer, by contrast, strictly preserves the standard Transformer computation graph and its overall computational footprint. By introducing geometric inductive biases without moving to heavier mathematical machinery, the model directly inherits the highly optimized hardware efficiency and the proven empirical scaling laws of the standard Transformer.
>
> We will incorporate the table related to the equivariance error and a new section detailing our remarks on the scaling potential of our Platonic transformer in the updated manuscript.
>
> We thank you again for your insightful comments and would appreciate your considering raising your score if you believe they have been addressed sufficiently, or following up with any questions we may help clarify further.

---

> > ### Author Rebuttal · Reviewer_uEBM · 2026-04-04
> >
> > I would like to thank the authors for answering my questions and for providing results on the equivariant error for different symmetry groups.
> >
> > My main concerns regarding the scalability claims of the method remain partial unresolved. I understand that the computational constraints can limit the ability to showcase a extensive scaling ablation. Nevertheless the current claim of improved scalability compared to more specialized equivariant networks is based mainly on the similarity of the proposed architecture to the transformers architecture. The comparable performance in the OMol25 benchmark provides a single data point, and is not sufficient to indicate a scaling trend. As such it is difficult to evaluate the correctness of the scalability claim fully, until a scaling analysis is performed.  Due to this limitation I will keep my recommendation of weak accept.

---

> > > ### Author Response · Authors · 2026-04-07
> > >
> > > Dear Reviewer, we thank you for this clarification. We agree that the OMol25 experiment provides only a single data point and does not, by itself, establish a full scaling trend. Our intention was not to overstate this evidence, but rather to show an initial indication that the proposed architecture remains competitive and efficient in a larger-scale regime. We therefore view the current result as a step in the right direction, while a systematic scaling analysis remains important future work. We will revise the manuscript to make this point more precise and better align the discussion with the current empirical evidence.

---

### Decision · Program_Chairs · 2026-04-30

**Decision:**

Accept (regular)

**Comment:**

The proposed  Platonic Transformer offers a clean and practical method for equivariance at zero additional computational cost, validated across six diverse tasks. Three of four reviewers support acceptance, and I agree that the contribution is potentially interesting and sufficient.
Required revision: Please add an experiment on a 3D task on point clouds (arbitrarily oriented ModelNet40, see for example "Vector Neurons: A General Framework for SO(3)-Equivariant Networks" ) where invariance is critical, as suggested by Reviewer iwnT. Please Include the other committed revisions as well.